# Active site geometry stabilization of a presenilin homolog by the lipid bilayer promotes intramembrane proteolysis

**Lukas P Feilen[1], Shu-Yu Chen[2], Akio Fukumori[3], Regina Feederle[1,4], Martin Zacharias[2], Harald Steiner[1,5]\***

[1]German Center for Neurodegenerative Diseases, Munich, Germany; [2]Center of Functional Protein Assemblies and Physics Department T38, Technical University of Munich, Garching, Germany; [3]Department of Pharmacotherapeutics II, Osaka Medical and Pharmaceutical University, Takatsuki, Japan; [4]Institute for Diabetes and Obesity, Monoclonal Antibody Core Facility, Helmholtz Munich, German Research Center for Environmental Health, Neuherberg, Germany; [5]Biomedical Center (BMC), Division of Metabolic Biochemistry, Faculty of Medicine, Munich, Germany

**Abstract** Cleavage of membrane proteins in the lipid bilayer by intramembrane proteases is crucial for health and disease. Although different lipid environments can potently modulate their activity, how this is linked to their structural dynamics is unclear. Here, we show that the carboxy-peptidase-like activity of the archaeal intramembrane protease PSH, a homolog of the Alzheimer's disease-associated presenilin/γ-secretase is impaired in micelles and promoted in a lipid bilayer. Comparative molecular dynamics simulations revealed that important elements for substrate binding such as transmembrane domain 6a of PSH are more labile in micelles and stabilized in the lipid bilayer. Moreover, consistent with an enhanced interaction of PSH with a transition-state analog inhibitor, the bilayer promoted the formation of the enzyme's catalytic active site geometry. Our data indicate that the lipid environment of an intramembrane protease plays a critical role in structural stabilization and active site arrangement of the enzyme-substrate complex thereby promoting intramembrane proteolysis.

**\*For correspondence:**
harald.steiner@med.uni-muenchen.de

**Competing interest:** The authors declare that no competing interests exist.

## Editor's evaluation

This work provides a strong contribution to our understanding of intramembrane proteolysis and in particular the subtle structural but significant influence of the lipid bilayer on proteolytic activity and coordination of the active site geometry.

## Introduction

Intramembrane proteolysis is a crucial cellular mechanism underlying many fundamental physiological processes (*Erez et al., 2009*; *Beard et al., 2019*). It is also involved in pathological conditions, most prominently in Alzheimer's disease (AD). Here, intramembrane cleavage within the transmembrane domain (TMD) of the amyloid precursor protein (APP) derived C99 substrate by γ-secretase results in the release of a variety of amyloid β-peptide (Aβ) species (*Steiner et al., 2018*). The longer forms, Aβ42 and Aβ43, are toxic to neurons and believed to trigger the onset of AD (*Selkoe and Hardy, 2016*). γ-Secretase is a membrane-embedded protein complex consisting of four components (*Yang et al., 2017*). The catalytic subunit presenilin is an aspartyl intramembrane protease (*Wolfe et al., 1999*; *Li et al., 2000*; *Steiner et al., 2000*; *Steiner et al., 1999*; *Kimberly et al.,*

**eLife digest** Cutting proteins into pieces is a crucial process in the cell, allowing several important processes to take place, including cell differentiation (which allows cells to develop into specific types), cell death, protein quality control, or even where in the cell a protein will end up. However, the specialized proteins that carry out this task, known as proteases, can also be involved in the development of disease. For example, in the brain, a protease called γ-secretase cuts up the amyloid-β protein precursor, producing toxic forms of amyloid-β peptides that are widely believed to cause Alzheimer's disease.

Proteases like γ-secretase carry out their role in the membrane, the layer of fats (also known as lipids) that forms the outer boundary of the cell. The environment in this area of the cell can influence the activity of proteases, but it is poorly understood how this happens.

One way to address this question would be to compare the activity of γ-secretase in the lipid environment of the membrane to its activity when it is entirely surrounded by different molecules, such as detergent molecules. Unfortunately, γ-secretase is not active when it is removed from its lipid environment by a detergent, making it difficult to perform this comparison. To overcome this issue, Feilen et al. chose to study PSH, a protease similar to γ-secretase that produces the same amyloid-β peptides but remains active in detergent.

When Feilen et al. mixed PSH with lipid molecules like those found in the membrane and amyloid-β precursor protein, PSH produced amyloid-β peptides including those that are thought to cause Alzheimer's. However, when a detergent was substituted for the lipid molecules this led to longer amyloid-β peptides than usual, indicating that PSH was not able to cut proteins as effectively. The change in environment appeared to reduce PSH's ability to progressively trim small segments from the peptides.

Computer modelling of the protease's structure in lipids versus detergent supported the experimental findings: the model predicted that the areas of PSH important for recognizing and cutting other proteins would be more stable in the membrane compared to the detergent.

These results indicate that the cell membrane plays a vital role in the stability of the active regions of proteases that are cleaving in this environment. In the future, this could help to better understand how changes to the lipid molecules in the membrane may contribute to the activity of γ-secretase and its role in Alzheimer's disease.

2000) present in the mammalian γ-secretase complexes as either presenilin-1 (PS1) or presenilin-2 variant (*Yu et al., 1998*; *Saura et al., 1999*). Mutations in PS1 are the major cause of familial AD (FAD) and cause an imbalance in the production of Aβ species that leads to relative increases of the longer forms over the normally major form Aβ40 (*Steiner et al., 2018*). Presenilins are evolutionary highly conserved proteins and related to the signal peptide peptidase (SPP) family of intramembrane proteases (*Ponting et al., 2002*; *Weihofen et al., 2002*). Ancestral precursors of presenilin and SPP exist in several archaea (*Torres-Arancivia et al., 2010*) and share key signature motifs including the protease family-defining GxGD active site motif (*Steiner et al., 2000*) with presenilin and SPP. The archaeal homolog from *Methanoculleus marisnigri* JR1 termed presenilin/SPP homolog (PSH) is capable of cleaving C99 and several other substrates (*Torres-Arancivia et al., 2010*; *Dang et al., 2015*; *Naing et al., 2015*; *Naing et al., 2018*). Similar to presenilin in the γ-secretase complex, PSH appears to cleave C99 in a sequential manner starting by initial ε-site cleavages between L49 and V50 (ε49) or T48 and L49 (ε48) followed by the release of various Aβ species from stepwise carboxy-terminal trimming cleavages (*Dang et al., 2015*; *Takami et al., 2009*). However, in contrast to presenilin, which requires complex formation with the other γ-secretase complex components for activity (*Takasugi et al., 2003*; *Edbauer et al., 2003*; *Kimberly et al., 2003*), PSH is active without accessory components. The crystal structure of PSH revealed first important insights into aspartyl intramembrane proteases showing that the two catalytic aspartate residues of the active site in TMD6 and TMD7 directly face each other and locate in a water-accessible cavity (*Li et al., 2012*). Subsequent cryo-electron microscopy (cryo-EM) structural analysis of γ-secretase showed that presenilin adopts a structure in the complex very similar to that of PSH (*Sun et al., 2015*) with nearly identical positions of the catalytic residues (*Bai et al., 2015b*). Further cryo-EM studies showed that binding of APP

and Notch substrates causes major conformational changes in both enzyme and substrate (*Zhou et al., 2019*; *Yang et al., 2019*). These led to an enzyme-substrate complex (E-S) with an extended TMD6 by formation of a new and stable TMD6a helix as well as a hybrid β-sheet between enzyme and substrate that causes unfolding of the ε-cleavage site region in the substrate (*Zhou et al., 2019*; *Yang et al., 2019*). Interestingly, formation of the TMD6a helix was also observed by cryo-EM upon inhibitor binding thus partially mimicking the substrate-bound state (*Bai et al., 2015a*; *Yang et al., 2021*).

The very similar structural folds of presenilin and PSH and the ability to cleave C99 in the TMD at the same sites as γ-secretase (*Torres-Arancivia et al., 2010*; *Dang et al., 2015*) make PSH an attractive model for the intrinsic protease activity of presenilin. To gain basic insights into the enzymatic workings of presenilin proteases, we thus set out to characterize the influence of two fundamentally different hydrophobic environments on the activity of PSH and asked if cleavage of C99 by the solubilized enzyme in detergent micelles would differ from a lipid-reconstituted state and if so, whether such differences could be correlated with the structural dynamics of this prototype presenilin protease or its E-S. Although the influence of lipids on the activity of presenilin and other intramembrane proteases is well documented (*Paschkowsky et al., 2018*), there are so far no studies in which biochemically determined activities of these proteases were linked with structural information that could explain how lipids, in particular a membrane bilayer environment, affect intramembrane protease structural dynamics and enzyme function. Since presenilins are not active in detergent micelles without lipids (*Zhou et al., 2010*), this critical question can however not be addressed for γ-secretase directly and requires a suitable model protease such as PSH. We found that detergent-solubilized PSH has a reduced carboxy-terminal trimming activity, that is processivity, compared to γ-secretase giving rise to an increased production of very long Aβ species such as Aβ46. Strikingly, the reconstitution of PSH into a lipid bilayer strongly promoted the protease processivity to shorter Aβ species such as Aβ38 highlighting the important role of the lipid membrane environment for intramembrane proteolysis. Furthermore, it enhanced the binding of a transition-state analog (TSA) γ-secretase inhibitor (GSI) affinity probe suggesting a more stable active site conformation in the lipid bilayer. These biochemical studies were accompanied by comparative modeling and molecular dynamics (MD) simulations to study the effect of detergent micelle and membrane lipid environment on substrate-bound PSH. In good agreement with the experimental data, the computational data suggest that the stabilization of TMD6a and the active site can explain the increased processivity and inhibitor binding in the membrane bilayer. Mutational analysis confirmed the assumed critical functional role of β-sheet and TMD6a corroborating the computational analysis of substrate-bound PSH. Collectively, these data provide insights into how structural adaptations occurring in response to changes in the hydrophobic environment from a micellar membrane mimetic to a real lipid bilayer translate into activity changes of an intramembrane protease. Moreover, with general implications for intramembrane proteolysis, they show how a lipid bilayer allows the formation of a stabilized active site geometry poised for substrate cleavage.

## Results

### PSH cleaves APP C99 to longer Aβ species

To get insights into the intrinsic protease activity of presenilin, we set out to further characterize the intramembrane cleavage of C99 by PSH (*Figure 1A*). Consistent with previous findings (*Dang et al., 2015*), n-dodecyl β-D-maltoside (DDM)-solubilized, His-affinity-purified PSH could cleave the C99-based APP C100-His$_6$ substrate (*Edbauer et al., 2003*) as demonstrated by the generation of the APP intracellular domain (AICD) and Aβ cleavage products (*Figure 1B*). Cleavage was inhibited by the TSA GSI L-685,458 (*Shearman et al., 2000*; *Figure 1B*) although much higher, micromolar concentrations were needed for efficient inhibition compared to those known for γ-secretase (*Li et al., 2000*; *Shearman et al., 2000*). Analysis of the Aβ profile using Tris-Bicine urea SDS-PAGE (*Figure 1C*) and MALDI-TOF mass spectrometry (*Figure 1D*, *Figure 1—figure supplement 1*) showed that Aβ40 and Aβ42 were the major Aβ species produced with a preference of Aβ42 over Aβ40. Interestingly, besides the increased generation of Aβ42 even longer Aβ species such as Aβ46 were relatively abundant. This suggests that PSH cleaves C99 at the same sites as γ-secretase but with reduced processivity.

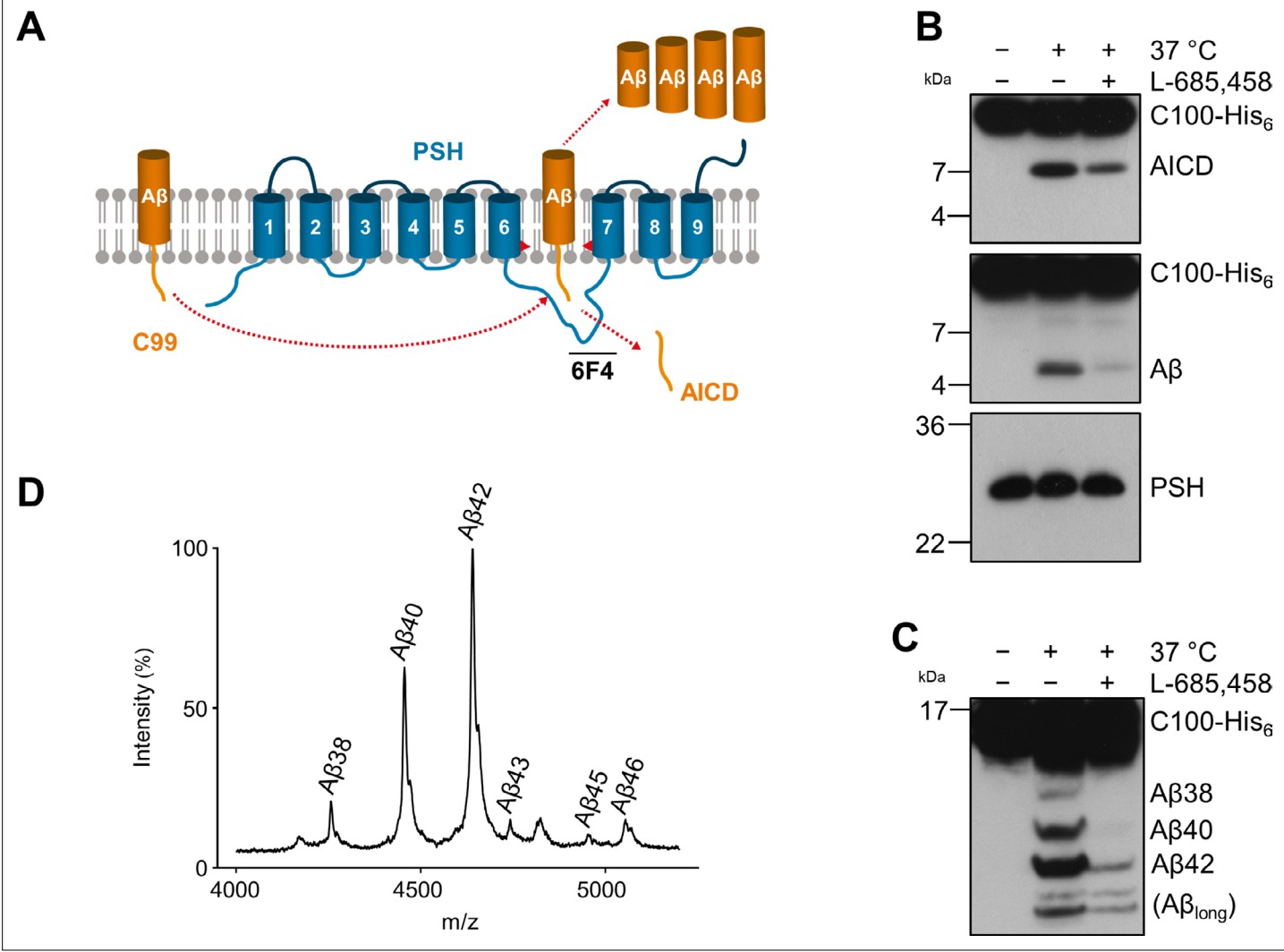

**Figure 1.** Cleavage of APP C99 by PSH. (**A**) Schematic illustration of APP C99 cleavage by PSH. PSH cleaves C99 and releases an AICD fragment and Aβ peptides. The epitope of the PSH specific antibody 6F4 in the loop between TMD6 and TMD7 is indicated. (**B**) Analysis of PSH activity in DDM micelles after incubation with C100-His$_6$ substrate overnight at 37 °C by immunoblotting for AICD (Y188) and Aβ (2D8). Specificity of substrate cleavage by PSH in the assay was controlled by sample incubation at 4 °C or 37 °C in presence of the GSI L-685,458 (20 µM). Immunoblotting of PSH (6F4) was performed to control for PSH levels. (**C**) Aliquot of samples from (**B**) separated by Tris-Bicine urea SDS-PAGE for identifying Aβ species produced by PSH in DDM micelles and analysis by immunoblotting (2D8). In (**B**) and (**C**), representative immunoblots from three to six independent biological replicates (i.e. independent protease preparations) are shown. (**D**) Representative MALDI-TOF MS spectrum of Aβ profile generated by PSH in DDM micelles from four independent biological replicates. The intensity of the highest peak was set to 100%. A GSI control is shown in *Figure 1—figure supplement 1* and observed masses for identified Aβ species are shown in *Figure 1—source data 1*.

The online version of this article includes the following source data and figure supplement(s) for figure 1:

**Source data 1.** Immunoblot images (raw and annotated) of cleavage assay (Source data for *Figure 1B, C*).

**Source data 2.** Calculated and observed masses for Aβ species in MALDI-TOF mass spectrometry (Source data for *Figure 1D*).

**Figure supplement 1.** MS specificity control for Aβ cleavage products generated from APP C99 by PSH.

## Lipid membrane enhances the processivity of PSH

We next investigated how a membrane environment of PSH influences the cleavage and processivity of the protease. Previous studies have shown that γ-secretase activity is dependent on the membrane environment as modulations of the lipid composition and/or bulk membrane properties in cell-free assays affected total activity as well as the ratios of the Aβ species generated (*Osenkowski et al., 2008*; *Osawa et al., 2008*; *Holmes et al., 2012*; *Winkler et al., 2012*). Furthermore, it was also shown that varying the pH in cell-free assays can modulate the total activity as well as the processivity of

γ-secretase (*Quintero-Monzon et al., 2011*). To investigate whether and how PSH cleavage of C99 would respond to a change from the micellar environment in DDM to a lipid bilayer environment, PSH was reconstituted in defined small unilamellar vesicles (SUVs) composed of palmitoyl-oleoyl PC (POPC), the most abundant phospholipid of biological membranes (*Figure 2—figure supplement 1A, B*). We then performed PSH in vitro assays with DDM-solubilized or POPC-reconstituted PSH in a pH range from 5.5 to 9.0. As shown in *Figure 2A*, the total activity was highest in the mild acidic to mild alkaline pH range and sharply dropped at pH values above 8.0. The pH optima for both conditions were very similar and lying around pH 7.0. However, compared to the DDM-solubilized enzyme, the processivity of PSH was strongly promoted in the lipid bilayer environment of the POPC SUVs as seen by a strongly increased production of Aβ38 and Aβ40 and the strong reduction of longer Aβ species (*Figure 2B*). Interestingly, the processivity of the reconstituted PSH appeared to be more reduced at alkaline pH values as judged from the appearance of Aβ species longer than Aβ42 (*Figure 2B*) at pH 7.5 and higher. A direct comparison of the Aβ profiles at pH 7.0 confirmed the increased processivity of PSH in the POPC lipid bilayer (*Figure 2C*, *Figure 2—figure supplement 2A*). Finally, we investigated the initial ε-site cleavages of C99 by PSH in DDM micelles or POPC vesicles at this pH. Mass spectrometry analysis showed that C99 was cleaved in both conditions at the ε49 and ε48 cleavage sites resulting in the release of the two N-terminally distinct AICD50 (ε49) and AICD51 (ε48) species (*Figure 2D*, *Figure 2—figure supplement 2B*). Collectively, these data show that the lipid environment increases the processivity of PSH in cleaving C99.

## Enhanced processivity of PSH is independent of the APP substrate N-terminus

Since the activities of DDM-solubilized and reconstituted PSH clearly differed, particularly in the processivity, we next sought to understand the underlying basis for this behavior at the level of its structural dynamics. Since there is no structure of PSH in complex with C99 available and structural investigations on γ-secretase in complex with an APP substrate were so far only performed with C83, an N-terminally shorter alternative C-terminal APP fragment generated by α-secretase (*Lichtenthaler et al., 2011*), we first tested whether C83 is processed similarly to C99. We thus analysed its cleavage by PSH in DDM micelles and the POPC bilayer at pH 7.0. C83 was cleaved by PSH in both conditions resulting in the generation of an AICD and the Aβ-equivalent cleavage product p3 (*Lichtenthaler et al., 2011*; *Figure 3A*). As judged from the processivity-reflecting ratios of p3 species ending at position 40 and 42, in contrast to the DDM micelle environment, the POPC bilayer enhanced the processivity and caused an increase in the relative production of shorter p3 species (*Figure 3B*). The increase in processivity was similar to that observed under these conditions for the corresponding Aβ species produced from C99 (*Figure 3C*). These data show that the increased processivity in the POPC environment is independent of the N-terminus of the substrate and that C83 and C99 behave comparable in both environments, so that C83 should be suitable as C99 surrogate for structural modeling and molecular dynamics simulations of PSH in complex with a substrate using information from the C83-bound γ-secretase.

## Structural modeling shows key features of substrate-bound γ-secretase in PSH

Since no experimental structure of the substrate-bound (holo) PSH is available, we generated 3 different starting structures, models 1, 2 and 3, for the PSH holo form using template-based modeling by assuming that PSH binds its substrate in a similar way as PS1. Since the PSH crystal structure (*Li et al., 2012*) (PDB 4HYG) misses several residues and loop segments in its substrate-free (apo) form, it is necessary to include the cryo-EM structure of holo PS1 bound to the C83 γ-secretase substrate (*Zhou et al., 2019*) (PDB 6IYC) as an additional template. In model 1, we used the entire holo PS1 including the C83 substrate as template whereas in models 2 and 3 most of the apo PSH structure was included and only different parts of missing loops segments were modeled based on the 6IYC template (including the C83 substrate, see Materials and methods for details). The overall structures of our holo PSH models are close to the experimentally determined apo PSH crystal structure (*Li et al., 2012*) but include several residues and loop segments that are missing in the apo crystal structure (*Table 1*, *Figure 4—figure supplement 1A, B*, *Figure 4—figure supplement 2*). In particular, as exemplified for model 2 (*Figure 4A*), it includes the TMD6a helix (residues H171-E177) C-terminal of

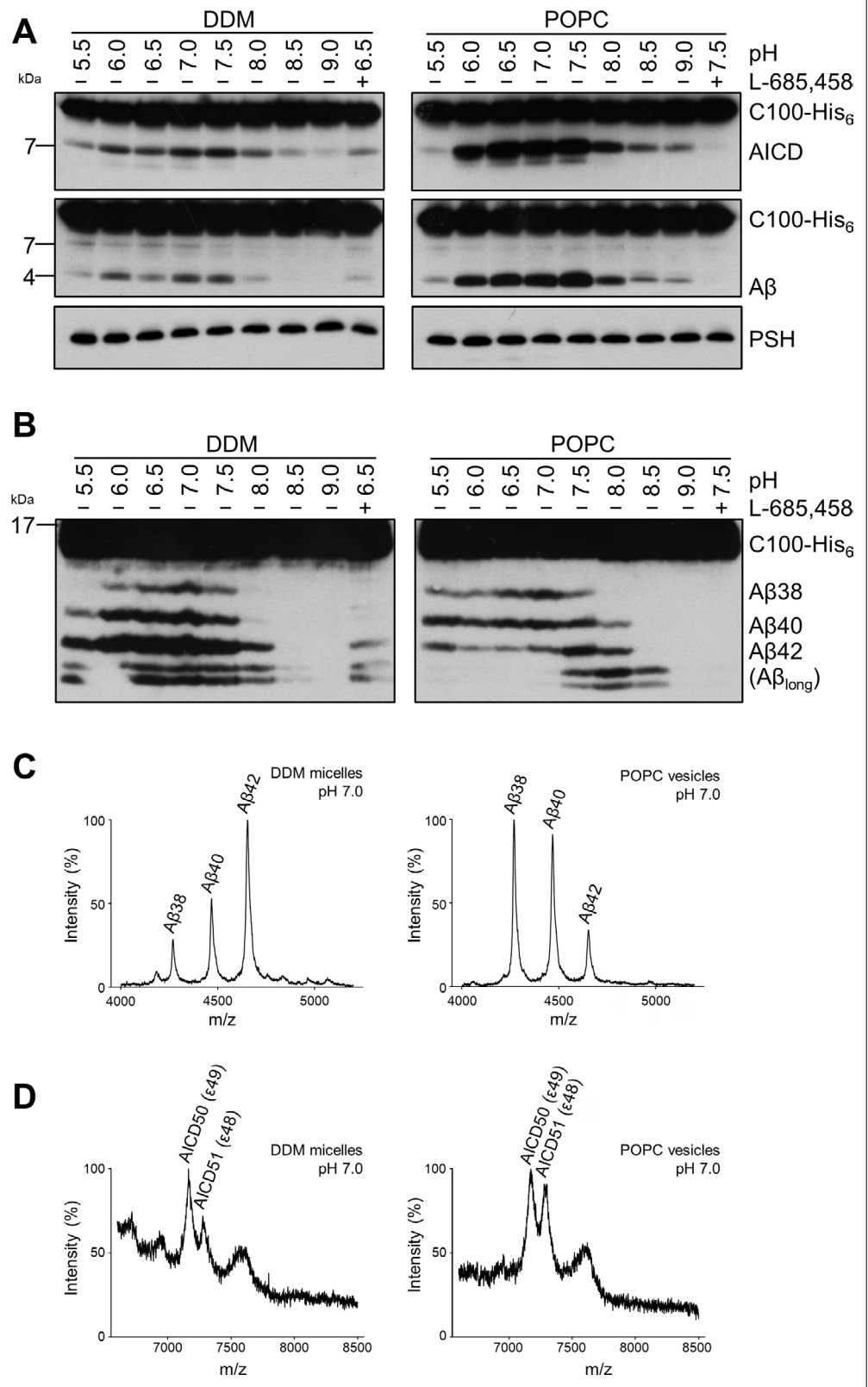

**Figure 2.** Comparison of PSH cleavage activity and processivity in DDM micelles and POPC bilayer. (**A**) Analysis of PSH activity in DDM micelles and POPC vesicles after incubation with C100-His$_6$ substrate at 37 °C overnight by immunoblotting for AICD (Y188) and Aβ (2D8). Immunoblotting of PSH (6F4) was performed to control for PSH levels. (**B**) Separation of Aβ species produced by PSH in DDM micelles and POPC vesicles by Tris-Bicine

*Figure 2 continued on next page*

*Figure 2 continued*

urea SDS-PAGE and analysis by immunoblotting for Aβ (2D8). In (**A**) and (**B**), representative immunoblots from six independent biological replicates are shown. Confirmation of PSH reconstitution in POPC SUVs is shown in *Figure 2—figure supplement 1*. (**C, D**) MALDI-TOF MS analysis of Aβ (**C**) and AICD (**D**) species generated by PSH in DDM micelles and POPC vesicles at pH 7.0. Representative mass spectra from four independent biological replicates are shown. The intensity of the highest peak was set to 100%. GSI controls are shown in *Figure 2—figure supplement 2* and observed masses for identified Aβ and AICD species are shown in *Figure 2—source data 1*.

The online version of this article includes the following source data and figure supplement(s) for figure 2:

**Source data 1.** Immunoblot images (raw and annotated) of cleavage assays (Source data for *Figure 2A, B*).

**Source data 2.** Calculated and observed masses for Aβ, and AICD species in MALDI-TOF mass spectrometry (Source data for *Figure 2C and D*).

**Figure supplement 1.** Reconstitution of PSH in POPC vesicles.

**Figure supplement 1—source data 1.** Raw values of measured fluorescence intensity (Source data for *Figure 2—figure supplement 1A*).

**Figure supplement 1—source data 2.** Immunoblot images (raw and annotated) of PSH reconstitution (Source data for *Figure 2—figure supplement 1B*).

**Figure supplement 2.** MS specificity controls for Aβ and AICD cleavage products generated from APP C99 by PSH.

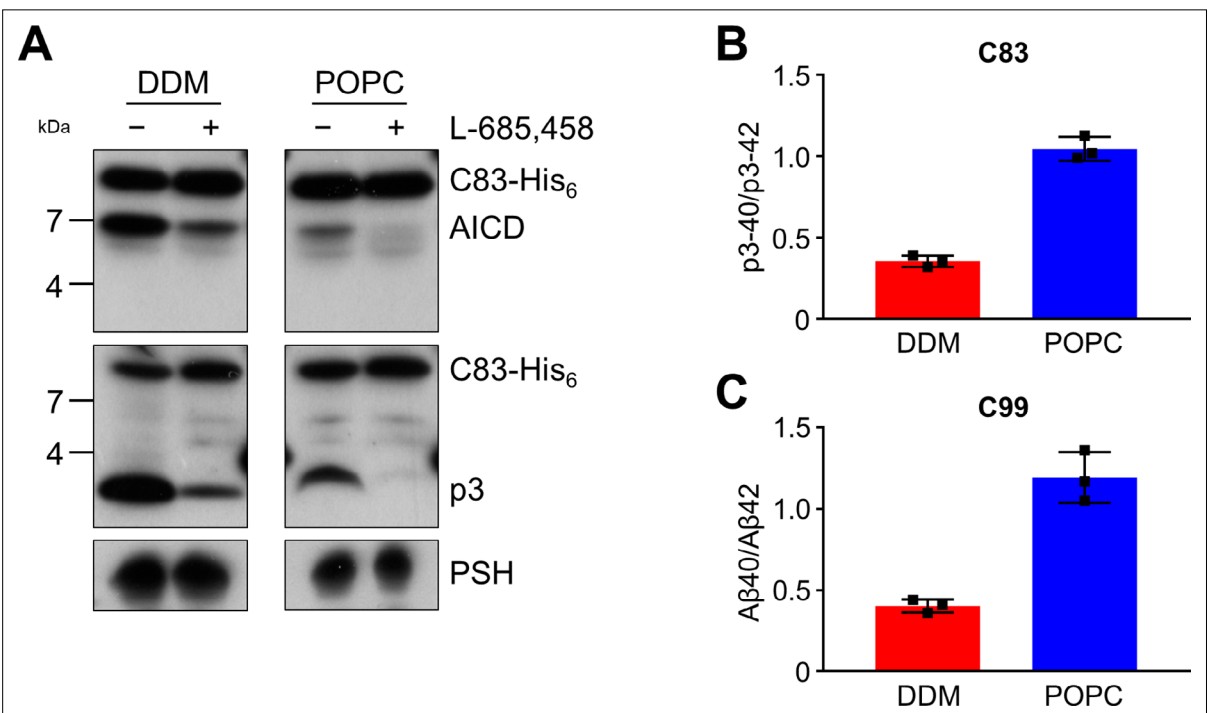

**Figure 3.** Cleavage of APP C83 by PSH. (**A**) Analysis of PSH activity in DDM and POPC environment after incubation with C83-His₆ and C100-His₆ substrates at 37 °C and pH 7.0 by immunoblotting for AICD (penta-His) and p3 (Aβ (22-35)). Immunoblotting of PSH (6F4) was performed to control for PSH levels. (**B, C**) p3-40/p3-42 ratio (**B**) and Aβ40/Aβ42 ratio (**C**) from PSH activity assays in DDM (red) and POPC (blue) environment analyzed by ECL-IA. Quantitative data are represented as mean ± standard deviation (SD) (n=3 biological replicates). Source data are shown in *Figure 3—source data 1*.

The online version of this article includes the following source data for figure 3:

**Source data 1.** Immunoblot images (raw and annotated) of cleavage assays (Source data for *Figure 3A*).

**Source data 2.** Raw values of p3 and Aβ concentrations measured in the ECL-IA and calculated p3-40/p3-42 and Aβ40/Aβ42 ratios (Source data for *Figure 3B, C*).

**Table 1.** Templates used for model building of PSH in complex with C83.
Residues of PSH used for model building are indicated.

| | Template | | |
| | 6IYC<br>(PS1, C83) | 4HYG<br>(Chain B) | Model 1 |
| --- | --- | --- | --- |
| Model 1 | * | – | – |
| Model 2 | – | L7-D162, D220-L292 | complete |
| Model 3 | – | L7-A176, E210-A293 | complete |

*PSH residues L7-R193 and E210-A293 were modeled based on the template.

TMD6 and the hybrid β-sheet formed between the β2-strand (A213-G217) and the β3-strand (V50-K54) of the substrate, which is stabilized by backbone interactions with residue Q272 preceding TMD9 (*Figure 4B*). These two structural elements were also found in the γ-secretase complexes with bound C83 (*Zhou et al., 2019*) or Notch1 (*Yang et al., 2019*) (PDB 6IDF) and were not present in the substrate-free γ-secretase complex. In addition, a salt bridge between R70 and E181 in the homology-modeled PSH (*Figure 4C*) replaced the hydrogen bond partners Y159 and R278 of the substrate-bound γ-secretase structure (*Figure 4D*). Thus, key features for substrate interaction (TMD6a and hybrid β-sheet) known from the γ-secretase–substrate complexes are analogously found in our models of the PSH–C83 complex.

## Verification of the β2-strand in the substrate-bound PSH models

Comparative modeling is sensitive to the choice of the templates and how the sequences are aligned together. A template with low quality or a sequence alignment with high uncertainties can lead to an unrealistic protein structure. Therefore, it is necessary to verify the presence of specific structural features in our substrate-bound PSH models that were observed in the substrate-bound PS1 structure. In case of γ-secretase, the deletion of the β2-strand (R377-L381) impaired the activity of the enzyme towards C83 and Notch1 (*Zhou et al., 2019*; *Yang et al., 2019*). To investigate whether this structural element is of similar functional importance in PSH, we mutated amino acid residues A213, F214, and V215 within the putative PSH β2-strand to prolines. Similar as done for PS1 (*Zhou et al., 2019*; *Yang et al., 2019*), we also deleted amino acid residues A213 to M216 and A213 to G217. When assessed for their enzymatic activity, the three mutants as well as the two deletion constructs showed clearly, and mainly strongly, decreased activities compared to wild type (WT) PSH in both DDM micelle or POPC bilayer conditions (*Figure 4E*). These results suggest that as the residues 213 to 216/217 are important for the activity of the protease they might indeed form the β2-strand observed in our structural models.

## Comparative molecular dynamics simulations of PSH in micelle and membrane environment reveals reduced PSH flexibility in the lipid bilayer

To get insight into the molecular details on how micelle and membrane environments might influence PSH conformational dynamics, the constructed C83-bound PSH models were embedded in DDM micelle (150 DDM molecules) or POPC bilayer (302 POPC molecules) environments and in each case three simulations with different distribution of starting velocities were performed (each simulation length: 0.6 μs). In total, six systems were constructed and 18 trajectories were generated in silico.

Snapshots of the PSH holo form in both environments are shown for model 2 in *Figure 5A*. All simulated systems stayed overall close to the starting structures with root-mean-square deviations (RMSD) relative to the start structure of about or less than 4.0 Å and an overall lower RMSD for all model 2 simulations (*Figure 5—figure supplement 1A*). Larger deviations were observed for all model 1 simulations and for one trajectory for model 3. It is likely that model 1 is less realistic than models 2 and 3 because it is entirely based on the PS1 template structure and is missing structural information from PSH.

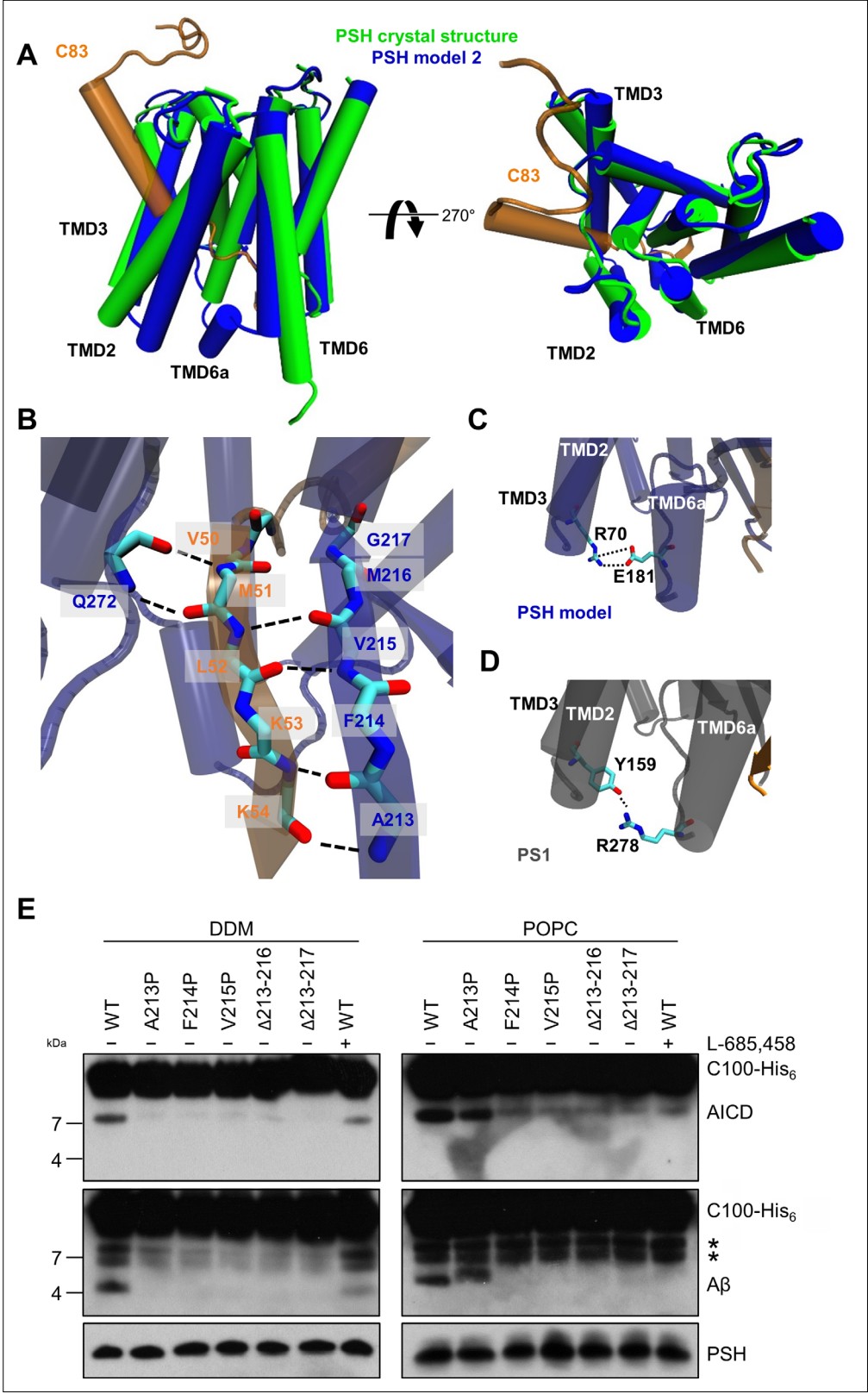

**Figure 4.** PSH homology model. (**A**) Alignment of the modeled holo form (model 2) of PSH (blue) with APP C83 substrate (orange) and the crystal structure of PSH (PDB 4HYG) in the apo form (green) in side view (left panel) and top view (right panel). An overlay of all three models and their RMSD values are provided in *Figure 4—figure supplement 1A* and B. (**B**) Schematic representation of the hydrogen bonds formed between β3 of the substrate

*Figure 4 continued on next page*

*Figure 4 continued*

(orange), β2, and Q272, respectively, of PSH (blue). (**C**) Interaction of TMD3 and TMD6a in the C83-bound PSH model through residues R70 and E181. (**D**) Interaction of TMD3 and TMD6a through residues Y159 and R278 in the C83-bound γ-secretase cryo-EM structure (PDB 6IYC). (**E**) Analysis of WT and mutant PSH activity in DDM and POPC environment after incubation with C100-His$_6$ substrate at 37 °C overnight by immunoblotting for AICD (Y188) and Aβ (2D8). Immunoblotting of PSH (6F4) was performed to control for PSH levels. The asterisks mark two substrate degradation bands, which are independent of PSH cleavage.

The online version of this article includes the following source data and figure supplement(s) for figure 4:

**Source data 1.** Immunoblot images (raw and annotated) of cleavage assays (Source data for *Figure 4E*).

**Figure supplement 1.** Comparison of the three different homology models.

**Figure supplement 2.** Alignment of PS1 and PSH used for homology modeling based on the TMD annotations of the available cryo-EM or crystal structures, respectively (PDB 6IYC for PS1 and PDB 4HYG for PSH).

We first analyzed the substrate mobility and interaction with PSH during the simulations in both environments. Stable substrate binding near the active site region involves the interaction of the β2-strand with the β3-strand at the C-terminus of the C83 substrate, which is required for substrate cleavage. The overall β-sheet interaction in terms of hydrogen bonds (H-bonds) was found to be similar in both the micelles and membrane environments except for model 3 where the H-bonds are more frequently formed in the bilayer (*Figure 5B*). We next investigated the mobility of the individual residues of C83 and PSH by calculating the root-mean-square fluctuation (RMSF) as well as the water accessibility of residues in the C83 substrate TMD by counting the average number of water molecules within 5 Å of the residue of interest. Both RMSF and water accessibility for each amino acid were not much different in both environments (*Figure 5C and D*). Notably, the substrate remained in a dry region from G37 to V46 and abruptly gained an increase in water accessibility around T48 and L49, which correspond to the initial ε-cleavage sites of C83 (and C99).

To quantitatively evaluate the flexible regions of PSH, RMSF profiles of each residue along the PSH sequence were calculated (*Figure 5E*). In all three models, larger fluctuations of hydrophilic loop 1 (HL1, between TMD1 and 2) were observed in DDM compared to POPC (*Figure 5E*). Similarly, the atomic fluctuations of TMD6a in models 1 and 2 were reduced in POPC versus DDM but no differences in the two environments were observed for TMD6a in model 3. In both environments, the atomic fluctuations of residues C-terminal of TMD6a stayed reduced in model 1 while they increased in model 3. In contrast to models 1 and 3, the atomic fluctuations of these residues stayed reduced in POPC and increased in DDM in model 2 (*Figure 5E*, *Figure 5—figure supplement 1B*). Furthermore, secondary structure analysis showed that TMD6a is mostly unfolded in model 1 and stable in model 3 (*Figure 5—figure supplement 2A, C*). Strikingly, when model 2 is placed in the micelle environment, TMD6a of substrate-bound PSH underwent a conformational transition between an α-helix and a loop structure (*Figure 5—figure supplement 2B*), while this transition was not observed for TMD6a in the POPC bilayer showing that the membrane environment stabilizes TMD6a (*Figure 5—figure supplement 2B*).

Despite the high structural similarity between the starting structures of model 2 and model 3 (RMSD = 0.163, *Figure 4—figure supplement 1B*), a difference in TMD6a positioning was observed during the simulations of these two models in the POPC environment. While TMD6a is located closer to C83 in model 2, it is located further away from C83 in model 3 (*Figure 5—figure supplement 3A*). In addition, when we directly compared the RMSF profiles of models 2 and 3 (*Figure 5E*) with each other, we observed that the residues immediately C-terminal of TMD6a are slightly more mobile in model 3 (*Figure 5—figure supplement 3B, C*). Because these residues are spatially close to HL4 between TMD4 and TMD5, HL4 also becomes more flexible in model 3 (*Figure 5—figure supplement 3B*). In fact, the conformational discrepancy in these regions arise from the model building of the two models which were built differently for residues A163 to R193 and E210 to G219 (*Table 1*, see Materials and methods for details). With a closer contact between TMD6a and C83 as well as an overall lower RMSD for model 2, we justified that model 2 may describe the dynamics of the PSH-C83 complex best so that model 2 is therefore used for our following analysis.

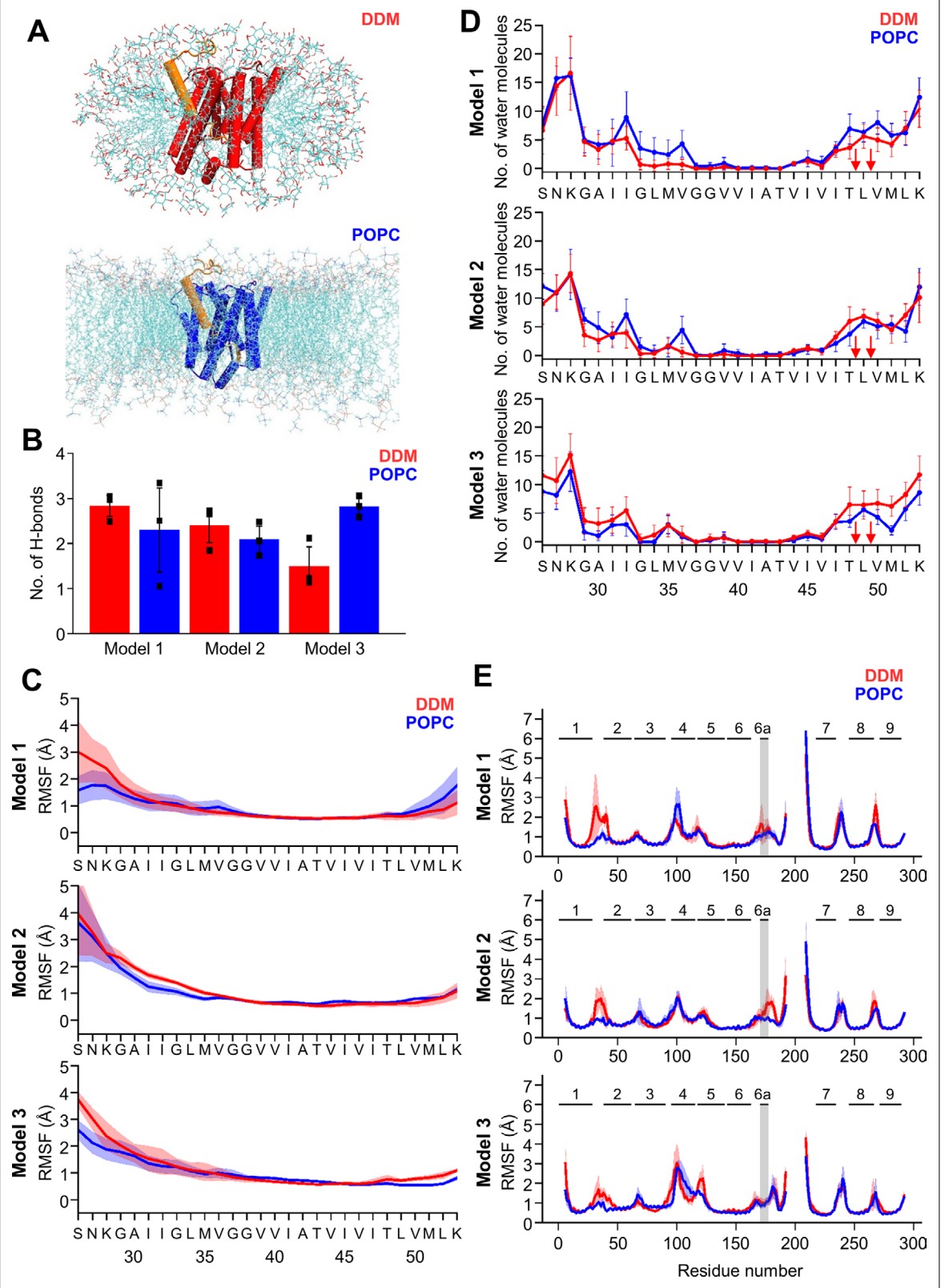

**Figure 5.** MD simulations of holo PSH forms in DDM micelle or POPC bilayer. (**A**) PSH with bound C83 substrate (model 2) embedded in a DDM micelle environment (upper panel) and a POPC bilayer (lower panel). (**B**) The average number of H-bonds formed between the β3-strand of C83 and the β2-strand of PSH. Each data point stands for the average value throughout one trajectory and the error bars represent the SD of the mean of three data points. (**C**) The backbone RMSF of C83 of different models in DDM (red) and POPC (blue) environments averaged over three trajectories. The

*Figure 5 continued on next page*

*Figure 5 continued*

shaded areas represent the SD of the mean. (**D**) Water accessibility along the substrate TMD residues extracted from the simulations of the holo PSH in DDM (red) and POPC (blue) environments (water accessibility for a residue is obtained as the mean number of water molecules within 5 Å of any atom of the residue). The red arrows indicate the position of the two ε-cleavage sites. The error bars represent the SD of the mean (n=3 trajectories). (**E**) The backbone RMSF of PSH of different models in DDM (red) and POPC (blue), environments averaged over three trajectories (note that residues 194–209 are not included in our PSH models). The gray boxes highlight TMD6a and the shaded areas represent the SD of the mean. Enlarged views on backbone RMSF of residues K170 to P185 (including TMD6a) are shown in *Figure 5—figure supplement 1B*.

The online version of this article includes the following source data and figure supplement(s) for figure 5:

**Source data 1.** Raw values of simulation data analysis (Source data for *Figure 5B–E*).

**Figure supplement 1.** Homology modeling of the holo form PSH in complex with C83 in DDM and POPC environments.

**Figure supplement 1—source data 1.** Raw values of simulation data analysis (Source data for *Figure 5—figure supplement 1A, B*).

**Figure supplement 2.** Secondary structure of PSH TMD6a and surrounding residues over simulation time.

**Figure supplement 3.** Comparison of models 2 and 3.

**Figure supplement 3—source data 1.** Raw values of simulation data analysis (Source data for *Figure 5—figure supplement 3B, C*).

## DDM insertion leads to an unwinding of PSH TMD6a

The biochemical cleavage data as well as the fluctuations in the MD simulations indicate a potential weakening of the E-S interaction in the micelle environment. This might explain the remarkable shift in processivity of PSH in the presence of a membrane lipid bilayer. In our PSH models, TMD6a creates a hydrophobic patch (formed by residues M172, I173, L175, and A176) that contacts the C83 substrate in the ε-cleavage site region (e.g. V50 and L52, *Figure 6A*). Similar interactions are found in the experimentally resolved C83-bound and Notch1-bound γ-secretase structures (*Zhou et al., 2019*; *Yang et al., 2019*), as well in the GSI-bound γ-secretase structures (*Yang et al., 2021*; *Figure 6—figure supplement 1A-D*). To gain a more detailed mechanistic view on how DDM and POPC molecules modulate the E-S stability, we calculated the structural properties of these molecules. Furthermore, we investigated how these molecules interact with PSH and C83. In the bilayer environment, the simulations indicate that POPC molecules are well ordered as indicated by a high lipid order parameter $S_{CH}$ (*Figure 6—figure supplement 2A, B*) and the computed area per lipid of ~68 Å² is close to the experimentally determined value of 64.3 Å² (*Kučerka et al., 2011*; *Figure 6—figure supplement 2C*). In contrast, in the micelle environment, DDM molecules are more mobile and can change their orientation more freely, as indicated by the lower lipid order parameter $S_{CH}$ for DDM compared to POPC (*Figure 6—figure supplement 2A, B*). In a larger micelle environment, with 50% more DDM molecules (225 DDM molecules, *Figure 6—figure supplement 3A*) the lipid order parameter of DDM improved (*Figure 6—figure supplement 3B*). Nevertheless, the number of H-bonds in the β-sheet (*Figure 6—figure supplement 3C*), the RSMF profile (*Figure 6—figure supplement 3D*) and the water accessibility in the C83 substrate (*Figure 6—figure supplement 3E*) did not differ from the respective values in the smaller DDM micelle (150 DDM molecules). The atomic fluctuations also did not differ largely between the different micelle sizes except for strongly reduced fluctuations of HL1 in the larger micelle (*Figure 6—figure supplement 3F*) originating from the more ordered DDM molecules around HL1 (*Figure 6—figure supplement 3H*). Furthermore, TMD6a and residues C-terminal of TMD6a were found to be also flexible in the larger DDM micelle as observed for the smaller micelle (*Figure 6—figure supplement 3F, G*), indicating that the size of the DDM micelle does not influence the observed differences in flexibility of TMD6a between DDM and POPC environment.

Although most of the DDM molecules are well aligned to the membrane normal at the protein periphery, some can transiently flip to a direction perpendicular to the membrane normal near to the gaps between TMDs (such as TMD2-TMD6 and TMD3-TMD4) (*Figure 6B–D*, *Videos 1 and 2*). When inserted into the intramolecular gaps between TMD2 and TMD6, the DDM molecule perturbs intramolecular interactions by forming unspecific hydrogen bonds with the adjacent amino acid backbones, and thus destabilizes TMD6a (*Figure 6B–C*). In addition, DDM inserts between TMD3 and TMD4 of PSH in the micelle environment and interacts with the loop C-terminal of TMD6a, corresponding to the higher RMSF observed for these residues (*Figures 5E and 6D*). In contrast, the well-ordered POPC molecules do not enter into the gaps between TMDs (*Video 3*) and do, therefore, not disturb intramolecular interactions. Collectively, these data suggest that a membrane lipid

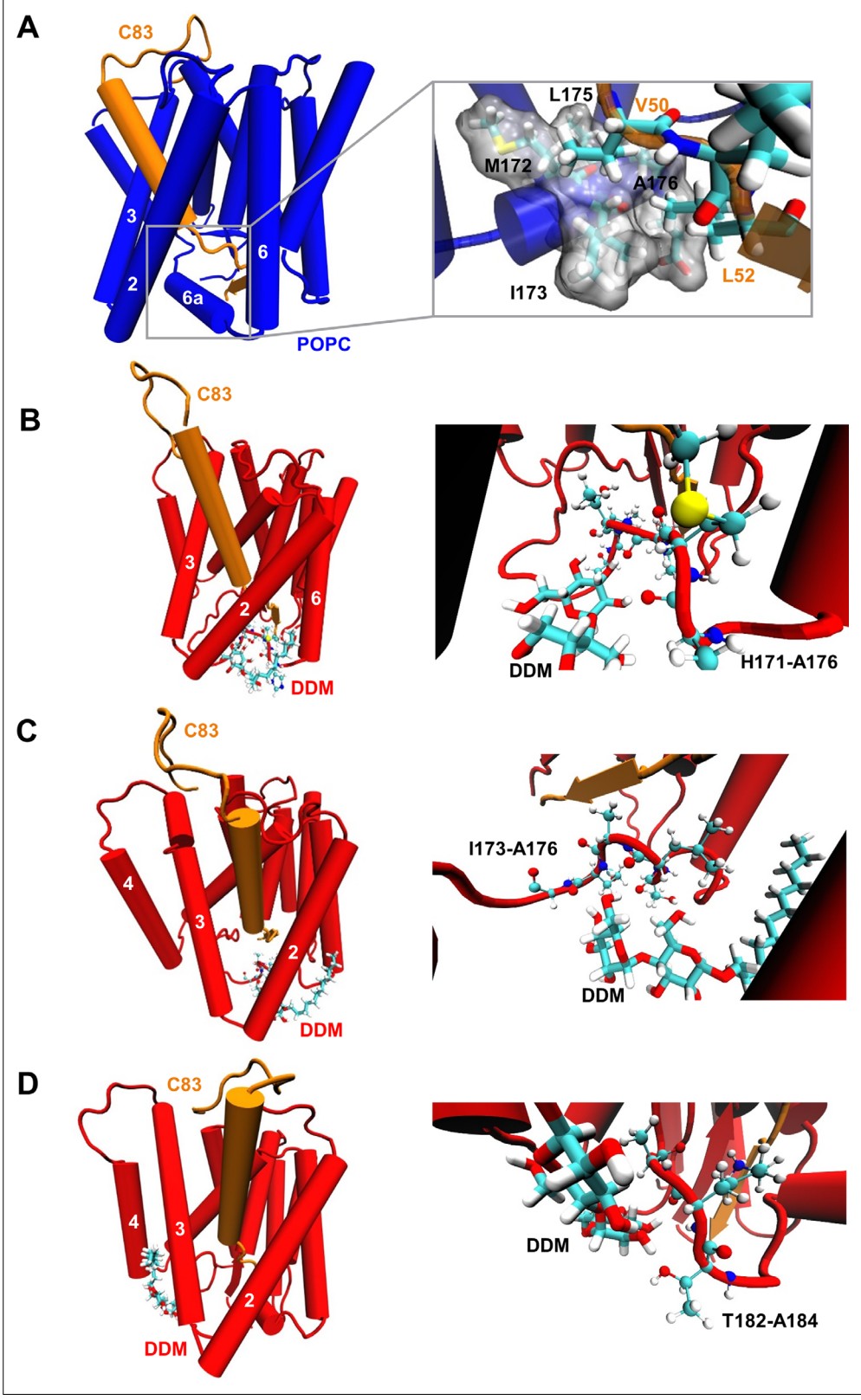

**Figure 6.** Destabilization of PSH TMD6a in a DDM micelle. (**A**) Hydrophobic interactions between PSH regions and the C83 substrate. The right panel shows an enlarged view of the interaction of a hydrophobic patch (gray surface) of TMD6a with V50 and L52 of the substrate in the POPC bilayer. (**B, C**) Snapshots of the DDM insertion between TMD2 and TMD6 in the first (**B**) and second (**C**) run of the simulations in DDM environment at 400 ns. The right

*Figure 6 continued on next page*

*Figure 6 continued*

panels show an enlarged view of the unspecific hydrogen bonding interactions between the DDM molecule and the TMD6a amino acid backbones. (**D**) Snapshot of the DDM insertion between TMD3 and TMD4 of the simulation in DDM environment at 430 ns. The right panel shows an enlarged view of the unspecific hydrogen bonding interactions between the DDM molecule and the amino acid backbones of the residues immediately C-terminal of TMD6a.

The online version of this article includes the following source data and figure supplement(s) for figure 6:

**Figure supplement 1.** Hydrophobic interactions of PS1 TMD6a in substrate-bound and GSI-bound γ-secretases.

**Figure supplement 2.** Biophysical properties of DDM and POPC.

**Figure supplement 2—source data 1.** Raw values of simulation data analysis (Source data for *Figure 6—figure supplement 2B, C*).

**Figure supplement 3.** MD simulations in different DDM micelles.

**Figure supplement 3—source data 1.** Raw values of simulation data analysis (Source data for *Figure 6—figure supplement 3B-G*).

environment promotes the formation of a stabilized E-S of PSH with the APP C83 substrate by the stabilization of TMD6a, an important structural element involved in substrate stabilization.

## Lysine mutations in TMD6a lead to helix unwinding and reduced activity

Our computational results suggest that the TMD6a helix plays an important role for substrate binding of PSH. Furthermore, the residues of the hydrophobic patch in TMD6a (M172, I173, L175, and A176) of PSH correspond to a homologous hydrophobic patch in the TMD6a of PS1 (L271, V272, T274, and A275), which is also affected by FAD mutations (*Steiner et al., 2018*). Some of them display a strong loss of function such as L271V and T274R (*Sun et al., 2017*) supporting the idea that TMD6a has an important function in substrate cleavage. To investigate the functional role that TMD6a plays in substrate stabilization, we performed additional MD simulations of in silico generated lysine mutations of M172, I173, L175, and A176 in the TMD6a hydrophobic patch in order to disrupt its nonpolar character. Because TMD6a is already unstable in DDM, the simulations were performed in the POPC environment only. The RMSD plots indicate that in most of the simulations, mutated PSH remained in an overall stable structure with RMSDs ~3.0 Å (similar to WT) (*Figure 7A*). In addition, no significant difference was found in C83 RMSF (*Figure 7B*), residue-wise water distribution (*Figure 7C*) and the hydrogen-bonding pattern of the β-sheet C-terminal of the ε-cleavage site (*Figure 7D*).

The PSH RMSF plots show that all mutations destabilize TMD6a to different degrees while having only smaller effects in other regions in comparison to WT (*Figure 7E*). It is worth noting that

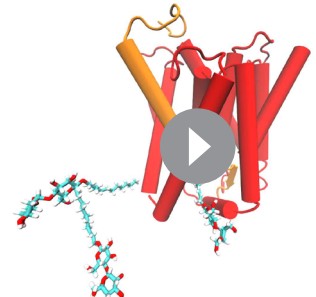

**Video 1.** Detergent-enzyme interaction in the DDM environment. 600 ns trajectories of PSH (red) in complex with C83 (orange) and nearby DDM molecules. A DDM molecule enters into the gap between TMD2 and TMD6. TMD6a switches between a helical and a loop conformation with the interference of the disordered DDM molecule.

https://elifesciences.org/articles/76090/figures#video1

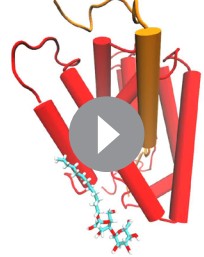

**Video 2.** Detergent-enzyme interaction in the DDM environment. 600 ns trajectories of PSH (red) in complex with C83 (orange) and nearby DDM molecules. A DDM molecule enters into the gap between TMD3 and TMD4. TMD6a switches between a helical and a loop conformation with the interference of the disordered DDM molecule.

https://elifesciences.org/articles/76090/figures#video2

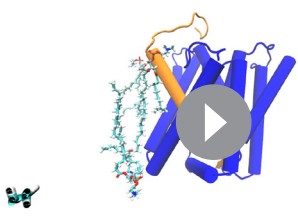

**Video 3.** Lipid-enzyme interaction in the POPC environment. 600 ns trajectory of PSH (blue) in complex with C83 (orange) and nearby POPC molecules. POPC molecules do not enter between TMD gaps and TMD6a remains a stable helix throughout the whole trajectory.

https://elifesciences.org/articles/76090/figures#video3

L175K introduces the largest TMD6a fluctuation, compared to the WT and the other three lysine mutants. Nevertheless, all four mutations distorted the helical structure of TMD6a in at least one of the simulations (*Figure 7—figure supplement 1*). Finally, to experimentally validate these structural predictions, we analyzed the cleavage of C99 by these mutant forms of PSH. All four mutants showed a strongly decreased, nearly abolished cleavage of C99 compared to WT PSH in both DDM micelle or POPC bilayer conditions (*Figure 7F*) suggesting that TMD6a and its hydrophobic patch is an important structural element of PSH.

## Lipid membrane environment stabilizes the active site geometry of PSH

The proteolysis reaction requires a specific geometry of all elements that form the active site. A critical issue is the distance between the two catalytic aspartic acids D162 and D220. Free energy calculation along the distance between D257 and D385 of PS1 has suggested that mutations disturbing the active site geometry and alter the distance between D257 and D385 correlate with changes in enzyme activity (*Chen and Zacharias, 2020*). In PSH, the geometry is characterized by the Cγ-Cγ distances between the D162 and D220. The distances appeared more frequently around 6.8 Å in our model when placed in a bilayer environment (*Figure 8A*, *Figure 8—figure supplement 1*). These distances correspond to a potentially catalytically active site geometry that can also accommodate a water molecule between the catalytic aspartates and L49 essential for the proteolytic cleavage (*Figure 8B*). In contrast, larger distances are more frequently sampled in the micelle environment. When the distance is enlarged here, an increased number of water molecules can access the catalytic center and disturb the catalytic geometry (*Figure 8B*). Proteolysis-compatible Cγ-Cγ distances below 7.0 Å were more frequently sampled in the membrane environment (~76%) compared to simulations in the DDM environment (~63%) (*Figure 8A*, *Figure 8—figure supplement 1*). Detailed geometries at the catalytic site of the E-S for a smaller and a larger Cγ-Cγ distance are depicted in *Figure 8—figure supplement 2A, B*.

To experimentally test whether the lipid environment influences the active site geometry of PSH, we used the L-685,458-based biotinylated affinity ligand Merck C (*Beher et al., 2003*) to capture PSH in DDM micelles and in POPC vesicles. As shown in *Figure 8C*, Merck C was able to capture specifically PSH as judged from binding competition in the presence of excess amounts of the parental compound L-685,458. In all these experiments, binding competition was stronger in the POPC vesicles than in DDM micelles indicating that the more labile DDM environment also weakens the competition of binding with the parental compound. In addition, also the background level of unspecific binding was higher in the latter environment contributing to the higher levels of unspecific PSH capture. In agreement with these observations, quantitation of specifically Merck C-bound PSH showed that the capture was markedly enhanced for the protease in the POPC bilayer (*Figure 8D*). Additional enzyme inhibition experiments further showed that both inhibitors, the parental L-685,458 as well as Merck C, inhibited PSH less well in DDM than in POPC (*Figure 8E and F*). All in all, these findings support the interpretation that the POPC bilayer stabilizes the active site, whereas it is destabilized in DDM micelles. Thus, these data suggest that the lipid environment stabilizes the geometry of the active site, which translates into the increased processivity of PSH in POPC vesicles.

Despite the improved binding and capture of PSH by the L-685,458-derived affinity probe in the POPC membrane environment, a potent inhibition of PSH by L-685,458 required rather high micromolar concentrations of this GSI. We thus finally tested whether other known GSIs would be more effective in inhibiting PSH cleavage of C99. Besides L-685,458, another TSA inhibitor (III-31C *Esler et al., 2002*, *Figure 8—figure supplement 3A, B*) and four non-TSA inhibitors with comparable potency (DAPT *Dovey et al., 2001*, LY411575 *Lanz et al., 2004*, Begacestat *Mayer et al., 2008*

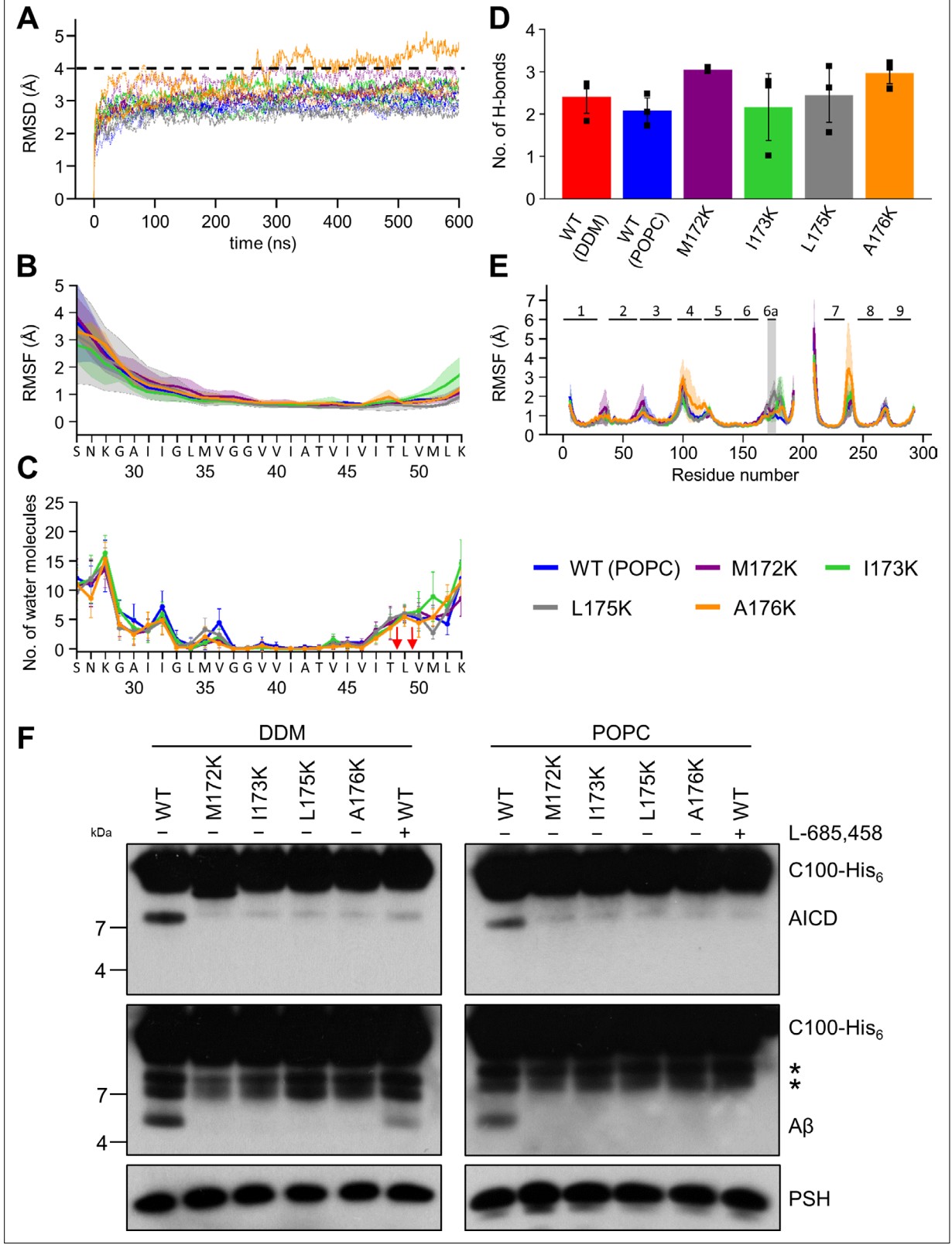

**Figure 7.** Impact of PSH TMD6a mutations on PSH structural dynamics and activity. (**A**) RMSD of the WT (blue) and the mutated systems M172K (violet), I173K (green), L175K (gray), and A176K (orange) in the POPC bilayer environment. The solid, dashed, and dotted lines represent three different simulations with random initial velocities. The black dashed line indicates an RMSD of 4 Å. (**B**) The backbone RMSF of C83 of WT PSH and different TMD6a lysine mutants in POPC environment averaged over three trajectories. The shaded areas represent the SD of the mean. (**C**) Water accessibility

*Figure 7 continued on next page*

*Figure 7 continued*

along the substrate TMD residues extracted from the simulations of the four lysine-mutant holo-form PSH systems in the POPC bilayer environments (water accessibility for a residue is obtained as the mean number of water molecules within 5 Å of any atom of the residue). The red arrows indicate the position of the two ε-cleavage sites. The error bars represent the SD of the mean (n=3 trajectories). (**D**) The average H-bond formed between the β3-strand at the C-terminus of C83 and the β2-strand of WT and lysine-mutated PSH. Each data point stands for the average value throughout one trajectory and the error bars represent the SD of the mean of three data points. (**E**) The backbone RMSF of WT (blue) and M172K (violet), I173K (green), L175K (gray), and A176K (orange) mutated PSH in POPC. The box highlights TMD6a and the shaded areas represent the SD of the mean (n=3 trajectories). Larger RMSF of the A176K mutant observed in residues 235–243 correspond to a folding-unfolding event in the mobile loop between TMD7 and TMD8 in the third trajectory (see *Figure 7—source data 1* and *Figure 7—source data 2*). (**F**) Analysis of WT and lysine-mutant PSH activity in DDM and POPC environment after incubation with C100-His$_6$ substrate at 37 °C overnight by immunoblotting for AICD (Y188) and Aβ (2D8). Immunoblotting of PSH (6F4) was performed to control for PSH levels. The asterisks marks substrate degradation bands, which are independent of PSH cleavage.

The online version of this article includes the following source data and figure supplement(s) for figure 7:

**Source data 1.** Raw values of simulation data analysis (Source data for *Figure 7A–E*).

**Source data 2.** Immunoblot images (raw and annotated) of cleavage assays (Source data for *Figure 7F*).

**Figure supplement 1.** Secondary structure of lysine-mutant PSH TMD6a and surrounding residues over simulation time.

and MRK-560 *Best et al., 2006*, *Figure 8—figure supplement 3C-F*) were tested for their potential to inhibit reconstituted PSH at pH 7.0. Remarkably, only the TSA inhibitor III-31C was able to inhibit C99 cleavage by PSH, whereas the non-TSA inhibitors were largely ineffective, even when used at the same high concentrations as for L-685,458 (20 μM) (*Figure 8G*). Overall, these inhibition data suggest that the stabilizing interactions of the TSA inhibitors in the PSH active site region may be different from that of γ-secretase (*Yang et al., 2021*; *Hitzenberger and Zacharias, 2019*). Such differences in stabilizing interactions might also affect the other GSIs that may bind too weakly to inhibit the enzyme.

Taken together, the MD simulations indicate a more stable, that is less fluctuating, geometry of the enzyme-substrate binding state around the enzyme active site in a POPC membrane environment compared to a DDM micelle environment. The DDM environment appears to destabilize important structural elements such as the TMD6a that is required for a stable enzyme-substrate interaction. These results can qualitatively explain the experimentally observed reduced processivity of PSH in DDM micelles and its boost in the POPC bilayer. Further, they emphasize the critical importance of the membrane environment for the formation of a conformationally stable active site geometry in the E-S complex, which is key for the efficient operation of intramembrane proteases in general.

## Discussion

It has previously been demonstrated that the archaeal intramembrane protease PSH cleaves the APP substrate C99 into Aβ40 and Aβ42 in a manner very similar to γ-secretase (*Dang et al., 2015*). PSH can thus be used as a surrogate for γ-secretase allowing to study the proteolytic activity of its catalytic presenilin subunit in the absence of its complex partners. Here, we confirm and extend these prior findings by a further, more in depth characterization of C99 processing by PSH. We first found that detergent-solubilized PSH cleaves C99 in DDM micelles with a reduced processivity as evident from higher amounts of Aβ42 than Aβ40. The reduced processivity of PSH under these conditions was supported by the identification of longer Aβ species such as Aβ46, cleavage products that were not identified in the previous study. Strikingly, we found in our assay system that the processivity was strongly enhanced in a membrane bilayer when the enzyme was reconstituted into POPC SUVs. Under these conditions, PSH processivity was strongly promoted as seen by the increased production of Aβ38. The protease was pH-dependent and showed the highest activity in the mild acidic to mild alkaline pH range in both micelle and bilayer conditions. In the POPC bilayer, the processivity of PSH was increased up to neutral pH before it rapidly dropped in the alkaline pH range of 7.5–8.5, where longer Aβ species started to accumulate eventually remaining unprocessed. Although the pH/activity profile of PSH showed overall similarities to that of γ-secretase, there were some notable differences. Compared to γ-secretase, which has a pH optimum of 6.5 (*Quintero-Monzon et al., 2011*), that of PSH was shifted to neutral pH. Moreover, although the processivity of γ-secretase was increasingly impaired toward more alkaline conditions like for PSH shown here, paradoxically, relative increases of Aβ38 were observed for γ-secretase in parallel (*Quintero-Monzon et al., 2011*). Clearly, the most

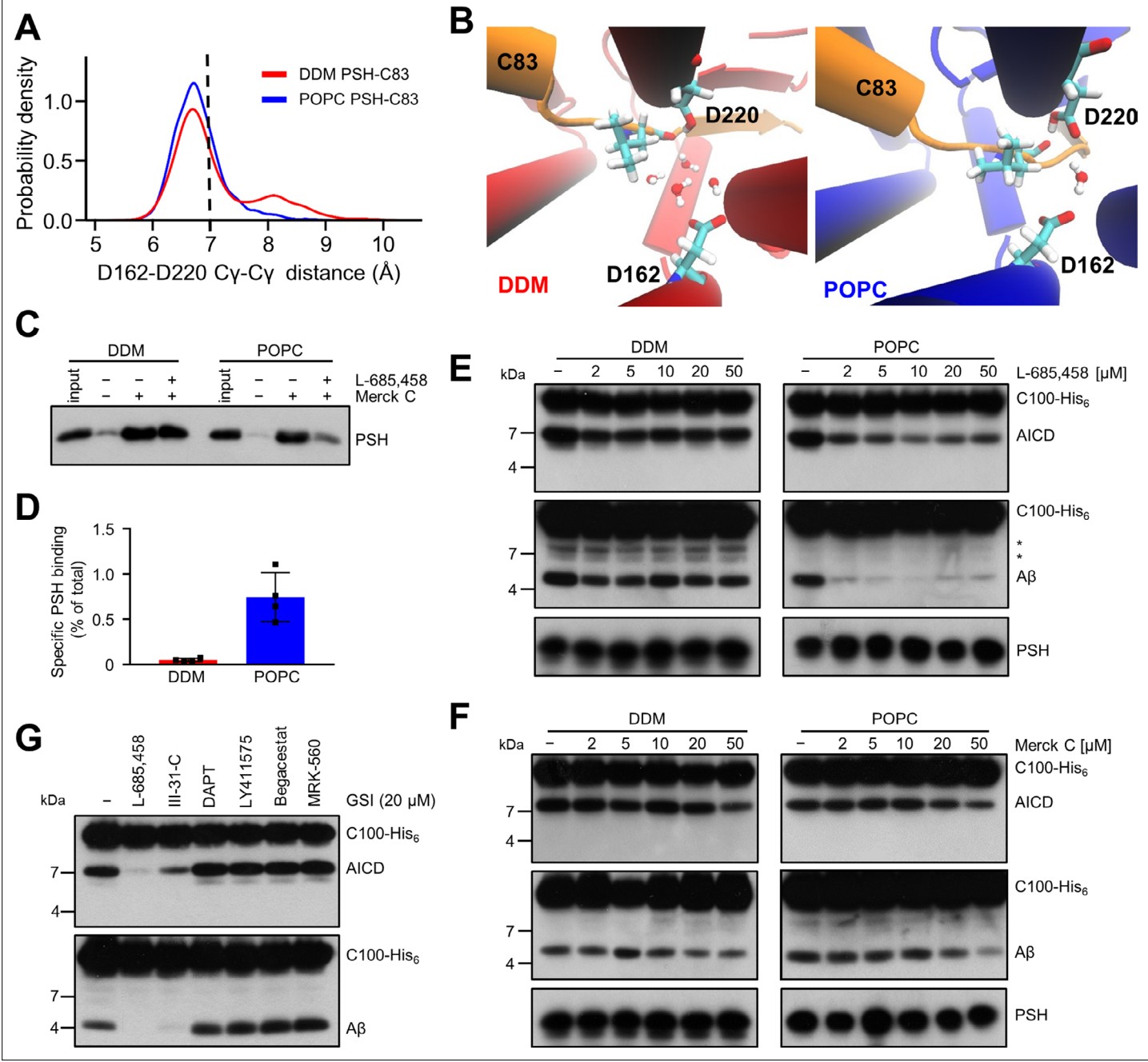

**Figure 8.** Stabilization of the PSH active site geometry in a POPC bilayer. (**A**) Histograms of the Cγ-Cγ distances between the D162 and D220 of PSH measured in DDM micelle (red) and POPC bilayer (blue) environments. The dashed line indicates the distance of 7 Å. The measured distances over time are shown in *Figure 8—figure supplement 1C*. (**B**) Snapshot of the catalytic cavity in DDM (left panel) and POPC (right panel) environment. The Cγ-Cγ distance between the two catalytic aspartates D162 and D220 is larger in DDM micelles and more water molecules enter the catalytic cavity between D162 and the substrate. Detailed geometries of these two active site conformations are depicted in *Figure 8—figure supplement 2*. (**C**) Immunoblot analysis of TSA-inhibitor binding to PSH in DDM micelles or POPC vesicles. PSH was affinity-precipitated by Merck C (a biotinylated derivative of L-685,458; 20 μM). To control for background binding and binding specificity, the affinity precipitation was assessed in the absence of Merck C as well as in the presence of excess amounts of the parental compound L-685,458 (2 mM) as competitor. The input represents 2.5% of the total sample used for the affinity precipitation. A representative immunoblot from four independent biological replicates is shown. (**D**) Quantitation of PSH binding by Merck C. Specific binding was defined as difference of PSH signals in the absence or presence of L-685,458 after additional subtraction of unspecific background binding signals. Quantitative data are represented as mean ± SD (n=4 biological replicates). The source data are shown in *Figure 8—source data 1*. (**E, F**) Inhibition assay of PSH in DDM micelles and POPC vesicles with increasing concentrations of L-685,458 (**E**) or Merck C (**F**), respectively. PSH activity was analyzed by immunoblotting for AICD (Y188) and Aβ (2D8) following incubation with C100-His$_6$ substrate at 37 °C overnight. Representative

*Figure 8 continued on next page*

*Figure 8 continued*

immunoblots from three independent biological replicates are shown. The asterisks mark two substrate degradation bands, which are independent of PSH cleavage. (**G**) Inhibition assay of PSH reconstituted in POPC vesicles in the presence of 20 µM TSA and non-TSA γ-secretase inhibitors. PSH activity was analyzed by immunoblotting for AICD (Y188) and Aβ (2D8) following incubation with C100-His$_6$ substrate at 37 °C overnight. Representative immunoblots from three independent biological replicates are shown.

The online version of this article includes the following source data and figure supplement(s) for figure 8:

**Source data 1.** Raw values of simulation data analysis (Source data for *Figure 8A* and *Figure 8—figure supplement 1A-C*).

**Source data 2.** Immunoblot images (raw and annotated) of inhibitor precipitation assay and cleavage assays (Source data for *Figure 8C, E-G*).

**Source data 3.** Raw values of immunoblot quantitation (Source data for *Figure 8D*).

**Figure supplement 1.** Cγ-Cγ distances in PSH homology model.

**Figure supplement 2.** Detailed geometry of the E-S catalytic site.

**Figure supplement 3.** Chemical structures of GSIs.

remarkable observation, however, was the rise in processivity when PSH was reconstituted into POPC membranes.

In searching for the underlying basis of these dramatic activity changes when changing from a micelle to a bilayer environment, we asked if these could be due to potential structural rearrangements that PSH undergoes in these two different environments. To investigate this possibility, PS1-based homology models were generated for PSH in the APP C83 substrate-bound holo form and the substrate-free apo form. The models revealed both β2-strand and TMD6a as structural elements, which we found by mutational analysis to be functionally highly critical for substrate cleavage by PSH. As observed for substrate-bound PS1 (*Zhou et al., 2019*; *Yang et al., 2019*), this suggests that they constitute important structural elements for substrate binding also for PSH.

The conformational dynamics of these structural models was evaluated in MD simulations to test whether structural changes might be observable that could explain the activity changes. We note that in previous simulations the atomistic dynamics of γ-secretase and the interaction with C83 have been studied (*Mehra et al., 2020*; *Bhattarai et al., 2020*), including also its activated state poised for ε-site cleavage (*Bhattarai et al., 2020*). However, comparative simulations in detergent micelles and lipid bilayer have so far not been performed. Among three models, model 2 is considered as the most realistic model and was chosen as working model for the E-S in our study. Our simulation results on PSH clearly showed more structural fluctuations of the protease in the micelle environment than in the membrane environment. These translated into less stable interactions with the substrate in the micelle compared to that in the bilayer, particularly of TMD6a with C83 in the active site region. In line with our mutational analysis, changing residues within the hydrophobic patch of TMD6a, which interacts with residues V50 and L52 at or near, respectively, the ε49-site of the substrate, disrupted interactions with the substrate in the MD simulations and strongly interfered with substrate cleavage in the PSH cleavage assays thus linking functional biochemical data with structural dynamics of PSH. TMD6a thus further emerges as an important structural element for substrate interaction that appears to be able to sense changes in the hydrophobic environment of the protease. The increased stability of TMD6a in POPC reflects a stabilized enzyme-substrate interaction that could likely translate into the enhanced processivity observed for the membrane environment. A more stable interaction increases the substrate residence time at the enzyme that has been shown for γ-secretase to be key for its processivity (*Okochi et al., 2013*) and that is also supported by MD simulations of a γ-secretase–C99 complex (*Dehury et al., 2019*). Moreover, a closer distance of the catalytic aspartate residues was much more frequently observed for the substrate-bound PSH holo form in the POPC bilayer suggesting that the formation of an active site geometry capable of peptide bond hydrolysis is promoted in the membrane environment. Consistent with these data, we found that binding of PSH to Merck C, a biotinylated derivative of L-685,458, was increased in the POPC bilayer. As shown previously for γ-secretase, L-685,458 interacts with the same subsite pockets as C83 and occupies a position of the substrate in the active site region close to where also the ε-cleavage sites of C83 become exposed and are unfolded (*Yang et al., 2021*; *Hitzenberger and Zacharias, 2019*).

Despite the demonstration of direct binding of the L-685,458 lead structure to PSH using the Merck C affinity ligand, L-685,458 inhibited PSH much less efficiently than γ-secretase, that is micromolar concentrations were needed to inhibit PSH compared to nanomolar concentrations known to

inhibit γ-secretase. Likewise, and consistent with previous results (*Dang et al., 2015*), the related TSA inhibitor III-31C could inhibit C99 cleavage of PSH but again at micromolar concentrations. Since other non-TSA GSIs failed to inhibit PSH, only TSA inhibitors can interact with PSH and effectively inhibit the enzyme. This suggests that the binding sites for non-TSA GSIs are different or, more likely, that their interactions with PSH are too weak to inhibit the enzyme. As shown previously, the binding sites of the non-TSA GSIs Avagacestat and Semagacestat are similar to the binding site of the TSA GSI L-685,458 (*Yang et al., 2021*). The non-TSA GSIs occupy the position of the β-strand of the substrate but do not protrude to the catalytic site resulting in decreased interactions with γ-secretase compared to the TSA GSI (*Yang et al., 2021*).

Both our experimental studies and the corresponding comparative MD simulations therefore suggest that the higher conformational flexibility of PSH in micelles causes destabilized interaction with C83 and C99 and consequently a reduced processivity. In contrast, a lipid bilayer induces a less flexible conformation of PSH that allows a more stable interaction with substrate, thereby promoting the processivity of PSH. Our data support recent findings for γ-secretase that showed differences in the processivity in a phospholipid/detergent-based versus a lipid raft-like membrane environment (*Szaruga et al., 2017*) and now provide an underlying molecular basis for this behavior. Similar to an artificial destabilization of the PSH/presenilin fold in detergent micelles, computational analyses suggest that FAD mutations in presenilin cause structural destabilizations (*Chen and Zacharias, 2020*; *Somavarapu and Kepp, 2016*; *Tang et al., 2019*) which are consistent with the experimentally observed impact on substrate interactions of these mutants (*Fukumori and Steiner, 2016*; *Trambauer et al., 2020*) and their alteration of APP/Aβ E-S stabilities resulting in processivity impairments (*Szaruga et al., 2017*). We also note that a less stable E-S in detergent micelles might account for differences in cleavage site usage and in inhibition profiles for diverse C99-based substrates that were observed in previous PSH assays (*Torres-Arancivia et al., 2010*; *Dang et al., 2015*; *Naing et al., 2018*).

Presumably, due to their non-native environment, the available structures of GxGD-type proteases show catalytically inactive conformations with too distant catalytic residues. The large distance of the catalytic aspartates of PSH in the substrate-free apo form is also seen for γ-secretase (10.6 Å, *Bai et al., 2015b*) as well as in different GxGD-type aspartyl proteases like FlaK (12 Å, *Hu et al., 2011*) and seems to represent their inactive form. Upon substrate interaction, this distance is decreased bringing the two catalytic aspartates closer to the initial cleavage sites (*Zhou et al., 2019*; *Yang et al., 2019*). As now shown in our study, a lipid bilayer environment promotes the formation of a stable active-site geometry by bringing the catalytic residues, water and the substrate scissile bonds into a conformation that allows proteolysis to proceed more efficiently. As a general implication for intramembrane proteolysis, our data suggest that a lipid bilayer-mediated stabilization of the active-site geometry might also be observable for other intramembrane proteases of different catalytic types.

Taken together, in good correlation between experimental and simulation data, our results with PSH as a model intramembrane protease highlight an important role of the membrane lipid environment in providing a stabilized E-S conformation that is crucial for substrate processing in intramembrane proteolysis. Our data further underscore a key role of the conformational flexibility of presenilin/PSH TMD6a for substrate interactions and proteolytic cleavage of presenilin-type proteases. Most importantly, they provide evidence that the lipid bilayer promotes the formation of a conformationally stable active site geometry, which is of general importance for an efficient catalytic operation of intramembrane proteases.

## Materials and methods

**Key resources table**

| Reagent type (species) or resource | Designation | Source or reference | Identifiers | Additional information |
|---|---|---|---|---|
| Strain, strain background (*Escherichia coli*) | BL21(DE3)$_{RIL}$ | Agilent Technologies | Cat# 230245 | |
| Recombinant DNA reagent | pQE60-C100-His$_6$ | *Edbauer et al., 2003* | N/A | |
| Recombinant DNA reagent | pQE60-C83-His$_6$ | This study | N/A | |

*Continued on next page*

*Continued*

| Reagent type (species) or resource | Designation | Source or reference | Identifiers | Additional information |
|---|---|---|---|---|
| Recombinant DNA reagent | pET21b-PSH | *Li et al., 2012* | N/A | Gift from Yigong Shi |
| Antibody | Anti-APP (C-terminus) Y188 (rabbit monoclonal) | Abcam | Cat# ab32136 | IB (immunoblot) (1:5000) |
| Antibody | Anti-APP (C-terminus) 6687 (rabbit polyclonal) | *Steiner et al., 2000* | N/A | IP (immunoprecipitation) (1:100–1:200) |
| Antibody | Anti-APP (Aβ22–35) Aβ (22-35) (rabbit polyclonal) | Sigma-Aldrich | Cat# A3356 | IB (1:1000) |
| Antibody | Anti-APP (Aβ1–16) 2D8 (mouse monoclonal) | *Shirotani et al., 2007* | N/A | IB (3 μg/ml) |
| Antibody | Anti-APP (Aβ17–24) 4G8 (mouse monoclonal) | BioLegend | Cat# 800702 | IB (1:500-1:2500) |
| Antibody | Anti-PSH (residues 192–204) 6F4 (rat monoclonal) | This study | N/A | IB (3 μg/ml), generation of antibody described further below |
| Antibody | Anti-Penta-His (mouse monoclonal) | Qiagen | Cat# 34660 | IB (1:1000) |
| Chemical compound, drug | Ni-NTA Agarose | Qiagen | Cat# 30210 | |
| Chemical compound, drug | Calbiosorb Adsorbent beads | Calbiochem | Cat# 206550 | Discontinued |
| Chemical compound, drug | POPC | Avanti Polar Lipids | Cat# 850457 P | Powder |
| Chemical compound, drug | Rhodamine-DHPE | Invitrogen | Cat# L1392 | |
| Chemical compound, drug | Sephacryl S-200 HR | GE Healthcare | Cat# 17058410 | |
| Chemical compound, drug | Streptavidin Sepharose | GE Healthcare | Cat# 17511301 | |
| Chemical compound, drug | L-685,458 | Sigma-Aldrich | Cat# 565771 | InSolution γ-Secretase Inhibitor X, used in cleavage assays |
| Chemical compound, drug | L-685,458 | Sigma-Aldrich | Cat# L1790 | Powder, dissolved in DMSO and used in inhibitor affinity precipitation experiments |
| Chemical compound, drug | Merck C | Taros Chemicals | N/A | Biotinylated L-685,458 |
| Chemical compound, drug | n-Dodecyl β-D-maltoside (DDM) | Millipore | Cat# 324355 | |
| Chemical compound, drug | Protein G Sepharose | Cytiva | Cat# 17061801 | |
| Chemical compound, drug | Protein A Sepharose | Cytiva | Cat# 17528001 | |
| Chemical compound, drug | Tropix I-BLOCK | Invitrogen | Cat# T2015 | |
| Chemical compound, drug | III-31C | Sigma-Aldrich | Cat# C0619 | |
| Chemical compound, drug | DAPT | Boehringer Ingelheim Pharma KG | N/A | |
| Chemical compound, drug | LY411575 | Karlheinz Baumann | N/A | |
| Chemical compound, drug | Begacestat | Karlheinz Baumann | N/A | |
| Chemical compound, drug | MRK-560 | Karlheinz Baumann | N/A | |
| Commercial assay or kit | V-PLEX Plus Aβ Peptide Panel 1 (4G8) Kit | Meso Scale Discovery | Cat# K15199G | |
| Commercial assay or kit | NativePAGE 4 to 16%, Bis-Tris, 1.0 mm, Mini Protein Gels, 10 wells | Invitrogen | Cat# BN1002BOX | |
| Software, algorithm | GelAnalyzer 19.1 | Istvan Lazar Jr., PhD Istvan Lazar Sr., PhD, CSc | N/A | http://www.gelanalyzer.com |
| Software, algorithm | AMBER18 | *Case et al., 2005* | N/A | |
| Software, algorithm | CHARMM-GUI | *Jo et al., 2008* | N/A | |

Continued

| Reagent type (species) or resource | Designation | Source or reference | Identifiers | Additional information |
|---|---|---|---|---|
| Software, algorithm | SWISS-MODEL | *Waterhouse et al., 2018* | N/A | |
| Software, algorithm | PROPKA3.1 | *Olsson et al., 2011*; *Sondergaard et al., 2011* | N/A | |
| Software, algorithm | DSSP | *Kabsch and Sander, 1983*; *Touw et al., 2015* | N/A | |

### Monoclonal antibody generation

Monoclonal antibody 6F4 (IgG2b/k) to PSH was raised in Wistar rat against amino acid residues 192–204 (KRADYSFRKEGLN) of PSH from *Methanoculleus marisnigri*.

### PSH constructs

All constructs are based on the PSH expression construct in pET-21b used for structure determination (*Li et al., 2012*). PSH point mutations and deletion were generated using site-directed mutagenesis. DNA sequencing of the newly generated plasmid confirmed successful mutagenesis.

### PSH expression and purification

Expression and purification of WT and mutant PSH was adopted from the published protocol (*Li et al., 2012*). In brief, *E. coli* BL21(DE3)$_{RIL}$ cells transformed with the pET-21b vector harboring an N-terminal 8 x His-tagged PSH were grown in LB medium to an optical density of 1.5 and expression was induced with 0.2 mM isopropyl β-D-1-thiogalactopyranoside (IPTG). PSH was expressed at 22 °C overnight and harvested cells were resuspended in resuspension buffer (25 mM Tris-HCl, pH 8.0, 150 mM NaCl). Cells were lysed by sonication; cell debris was removed by centrifugation and membranes were collected by ultracentrifugation at 150,000 x *g* for 1 hr. Membranes were solubilized in resuspension buffer containing 2% DDM by rocking at 4 °C for 2 hr. After ultracentrifugation at 150,000 x *g* for 30 min, the supernatant was incubated with Ni-NTA agarose beads (Qiagen) for 2 hr at room temperature. Beads were then washed with resuspension buffer containing 20 mM imidazole and 0.6% (w/v) DDM. PSH was eluted with resuspension buffer containing 250 mM imidazole and 0.6% (w/v) DDM. Correct folding of WT and mutant PSH was confirmed by dynamic light scattering (DLS, Malvern Instruments High Performance Particle Sizer) (*Appendix 1—figure 1A*), Blue Native (BN)-PAGE (*Appendix 1—figure 1B*) and nano differential scanning fluorimetry (nanoDSF, NanoTemper Tycho) (*Appendix 1—figure 1C*). For DLS, protein samples (25 µM) were analyzed in a Hellma Analytics High Precision Cell. For BN-PAGE, samples were prepared as described (*Schägger and von Jagow, 1991*) and separated using a Novex NativePAGE 4–16% Bis-Tris gel. Following electrophoresis, the gel was prepared for blotting as described (*Winkler et al., 2009*) and WT and mutant PSH were subjected to immunoblot analysis with antibody 6F4. For nanoDSF, protein samples (25 µM) were loaded into NanoTemper Tycho NT.6 capillaries, unfolding profiles of WT and mutant PSH were recorded, and the inflection temperatures ($T_i$) were obtained by automated data analysis.

### PSH reconstitution in POPC vesicles

PSH reconstitution into POPC SUVs was based on the reconstitution of γ-secretase into SUVs (*Winkler et al., 2012*). SUVs were prepared in a low citrate buffer (5 mM sodium citrate, 3.5% glycerol, pH 6.4) and diluted 2.5 times with buffer (5 mM sodium citrate, 3.5% glycerol, 30 mM DTT, pH 6.4). One volume of purified PSH and four volumes of the vesicle preparation were mixed in the presence of an excess of Calbiosorb adsorbent beads (Calbiochem) and incubated at 4 °C overnight to allow the formation of proteoliposomes.

### Validation of PSH reconstitution into POPC SUVs

To validate the incorporation of PSH into POPC vesicles, PSH was reconstituted into POPC vesicles containing the fluorescent marker lipid rhodamine-DHPE. These vesicles were then subjected to a small gel filtration column packed with Sephacryl S-200 HR to separate vesicles and free PSH. The vesicle content of each fraction was analyzed by measuring fluorescence ($\lambda_{ex}$ 530 nm, $\lambda_{em}$ 590 nm)

with Fluoroscan Asket Fl (Labsystems) and the presence of PSH in the fractions was analyzed by immunoblotting with antibody 6F4.

## APP substrate constructs

Recombinant APP substrate C100-His$_6$ was described before (*Edbauer et al., 2003*). The corresponding C83-His$_6$ (containing an N-terminal methionine) was generated by PCR and cloned into pQE60.

## Expression and purification of APP-based substrates

C100-His$_6$ and C83-His$_6$ were expressed in *E. coli* BL21(DE3)$_{RIL}$ cells after induction with IPTG at 37 °C for 4 hr. Cell pellets were resuspended in TE buffer (20 mM Tris (pH 7.5), 1 mM EDTA), sonified and inclusion bodies were collected by centrifugation. Inclusion bodies were lysed overnight at 4 °C in 20 mM Tris (pH 8.5), 6 M urea,1 mM CaCl$_2$, 100 mM NaCl, 1% (w/v) SDS and 1% (v/v) Triton X-100 by rotation. The lysate was diluted 1:5 with 20 mM Tris (pH 7.5) and 150 mM NaCl and then incubated with Ni-NTA agarose for 2 hr at room temperature. Ni-NTA beads were washed extensively with TX-wash buffer (50 mM Tris (pH 8.5), 300 mM NaCl, 1% (v/v) Triton X-100), SDS-wash buffer (50 mM Tris (pH 8.5), 300 mM NaCl, 0.2% (w/v) SDS) and imidazole wash buffer (50 mM Tris (pH 8.5), 300 mM NaCl, 0.2% (w/v) SDS, 20 mM imidazole) before the elution of bound protein with elution buffer (50 mM Tris (pH 8.5), 300 mM NaCl, 0.2% (w/v) SDS, 150 mM imidazole).

## PSH in vitro assay

The in vitro assays using recombinant APP substrates C100-His$_6$ and C83-His$_6$ were performed with either DDM-solubilized PSH or reconstituted PSH. To test PSH activity over a range of different pH values, the in vitro assay was performed in the presence of a master buffer (20 mM Bicine, 20 mM HEPES, 20 mM MES) adjusted to different pH values (pH 5.5–9.0). 1–2 μM PSH was incubated with 0.5 μM substrate overnight at 37 °C. The final DDM concentration was 0.02% for the assays in DDM micelles. For the assays with PSH reconstituted in POPC vesicles a small amount of DDM was added just below the critical micelle concentration (CMC) of 0.0087%. As for γ-secretase, detergent addition below the CMC is necessary to achieve enzyme activity after reconstitution (*Winkler et al., 2012*). Generated AICD, Aβ and p3 were analyzed by immunoblotting and in addition, Aβ and AICD species were determined by MALDI-TOF mass spectrometry (MS) analysis as described previously (*Winkler et al., 2009*; *Page et al., 2008*; *Ebke et al., 2011*). In brief, samples were diluted with IP-MS buffer (10 mM Tris (pH 8.0), 140 mM NaCl, 0.5 mM EDTA, 0.1% n-octyl-glucopyranoside) and immunoprecipitated for 16 hr at 4 °C with antibody 4G8 and protein G Sepharose for Aβ species or with antibody 6687 and protein A Sepharose for AICD species.

## Electrochemiluminescence immunoassay (ECL-IA)

Aβ and p3 species from PSH in vitro assays were analyzed with the V-PLEX Plus Aβ Peptide Panel 1 (4G8) Kit (Meso Scale Discovery, K15199G) using SULFO-tagged anti-Aβ antibody (4G8) in a 1:50 dilution. Samples were centrifuged for 30 min at 21,000 x *g* and then diluted 1:25 to reach a sample concentration in the linear detection range. The ECL-IA was performed following the manufacturer's protocol.

## Inhibitor affinity precipitations of PSH

Streptavidin Sepharose beads (GE Healthcare) were washed three times with PBS and then blocked overnight with 1% I-Block in PBS at 4 °C and additionally for 30 min at room temperature. PSH in DDM micelles or in POPC vesicles was diluted with MES-buffer (50 mM MES pH 6.0, 500 mM NaCl, 5 mM MgCl$_2$, 1 x PI mix complete (Roche)) to 1–2 μM PSH. To mimic the cleavage assay condition, a small amount of DDM was added just below the CMC of DDM (0.0087%) to the POPC reconstituted PSH. The diluted PSH solution was precleared with streptavidin beads for 30 min at 4 °C. To capture PSH, the precleared solution was incubated with 20 μM L-685,458-based biotinylated TSA inhibitor Merck C (*Beher et al., 2003*) (Taros Chemicals) in the presence of streptavidin beads for 2 hr at room temperature. To analyze non-specific binding, Merck C was omitted or a 100-fold molar excess of the parental compound L-685,458 was added. To quantify Merck C GSI binding to PSH, the chemiluminescence signal of the respective immunoblots were quantified using the LAS−4000

image reader (Fujifilm Life Science) and GelAnalyzer 19.1 software (http://www.gelanalyzer.com). Specific binding was calculated as the difference between the binding of PSH and the binding of PSH in presence of the competitor L-685,458 after subtraction of unspecific PSH binding to the beads.

## Molecular dynamics simulations

The available crystal structures of PSH, PDB 4HYG (*Li et al., 2012*) and PDB 4Y6K (*Dang et al., 2015*) include several amino acid substitutions compared to the WT PSH sequence and in addition, several important loop segments are missing. In both crystals, the enzyme forms a tetramer that may also stabilize a structure different from the solution and substrate-bound conformation. Under the assumption that a substrate is bound to PSH in an analogous fashion as in the PS1 homolog, we used the option to generate a comparative model structure using the C83 substrate-bound γ-secretase structure (PDB 6IYC) (*Zhou et al., 2019*) as a template. Three models were generated using the SWISS online server (*Waterhouse et al., 2018*) and the program MODELLER (*Eswar et al., 2006*) with a same sequence alignment strategy (*Figure 4—figure supplement 2*) but different homology protocols. Model 1 was built by taking only PDB 6IYC as the template as generated by the SWISS online server (*Table 1*). Model 2 was generated by using residues ranging from L7 to D162 and D220 to L292 from chain B of PDB 4HYG and model 1 as templates with the MODELLER multi-template method (*Table 1*).

Table 2. pK$_a$ predictions as calculated by PROPKA3.1 for published PSH and γ-secretase structures.

pKa values of the catalytic aspartate residue which is most likely protonated are indicated in red.

| PDB ID | Enyzme | Ligand | pKa (D162, D220) |
|--------|--------|--------|------------------|
| 4HYG | PSH | None | 5.04, 6.62 |
| 4Y6K | PSH | III-31-C | 5.63, 7.52 |

| PDB ID | Enyzme | Ligand | pKa (D257, D385) |
|--------|--------|--------|------------------|
| 4UIS | PS1 | None | 3.18, 6.16 |
| 5A63 | PS1 | None | 4.42, 6.16 |
| 5FN5 | PS1 | None | 4.98, 3.63 |
| 5FN4 | PS1 | Unknown helix | 4.70, 4.71 |
| 5FN3 | PS1 | Unknown helix | 4.90, 7.13 |
| 5FN2 | PS1 | DAPT | 5.13, 9.93 |
| 6IYC | PS1 | C83 | 6.39, X* |
| 6IDF | PS1 | Notch1 | 6.21, X* |
| 6LR4 | PS1 | Semagacestat | 6.12, 7.94 |
| 6LQG | PS1 | Avagacestat | 6.08, 7.22 |
| 7V9I | PS1 | L-685,458 | 7.11, 8.90 |
| 7D8X | PS1 | L-685,458 and E2012 | 7.01, 8.69 |

*For structure determination D385 was mutated to alanine and therefore no pKa value is given.

Model 3 was built by taking all residues resolved in chain B of PDB 4HYG and model 1 with the MODELLER multi-template method (*Table 1*). Similar to the PS1 template structure, the final holo PSH structure is composed of two fragments with an N-terminal fragment from L7-R193 and a C-terminal fragment from E210 to A293. The generated holo-state PSH structures were then embedded in two different environments: micelle capsules consisting of 150 DDM molecules and a membrane bilayer with 302 POPC molecules. An additional, larger micelle capsule system of model 2 was constructed with 50% more, namely 225, DDM, molecules. Lysine mutations M172K, I173K, L175K and A176K were constructed based on model 2 with RMSD (WT vs mutant) < 0.1 Å, and embedded in a membrane bilayer system with 302 POPC molecules using the CHARMM-GUI online server (*Cheng et al., 2013*; *Wu et al., 2014*). All 11 systems were prepared and solvated in explicit TIP3P water (*Jorgensen et al., 1983*) at a salt concentration of 0.15 M KCl using the CHARMM-GUI online server .

The interaction of proteins, lipid, and micelles is described by the charmm36m force field (*Huang et al., 2017*). Each system was simulated using the AMBER18 pmemd GPU accelerated version (*Case et al., 2005*) in combination with a Berendsen barostat (1 bar) and a Langevin thermostat (303.15 K). The hydrogen mass repartitioning method was used allowing a time step of 4 fs. Three simulations with 600 ns each were performed for each system, in total 33 NPT trajectories were generated for further analysis. Non-bonded cutoff was set to 12 Å with a force-based switching distance of 10 Å. D220 was selected to be protonated while D162 was unprotonated according to the pK$_a$ prediction on the existing PSH and PS1 structures by PROPKA3.1 (*Olsson et al., 2011*; *Sondergaard et al., 2011*; *Table 2*).

The lipid tail order parameter $S_{CH}$ was computed in model 2 in DDM and POPC environments to show the orientation and the ordering of the concerning CH vector (*Tieleman et al., 1997*; *Vermeer et al., 2007*) with respect to the protein principle axis, which was aligned to the lipid normal in the POPC environment. In addition, the area per lipid was computed in model 2 on both leaflets to verify the reliability of our POPC lipid model.

RMSF of PSH and C83 curves were calculated by taking the last 200 ns with the time-average PSH structure of each simulation as the reference and only taking the backbone atoms for the calculation. Secondary structure of PSH TMD6a was calculated using the DSSP method (*Kabsch and Sander, 1983*; *Touw et al., 2015*).

## Acknowledgements

We thank Yigong Shi for the PSH expression plasmid, Karlheinz Baumann for GSIs and Gabriele Basset, Frits Kamp and Alice Sülzen for technical assistance as well as Shibojyoti Lahiri, Ignasi Forné and Axel Imhof from the Protein Analysis Unit of the Biomedical Center Munich (BMC) and Michaela Smolle from the Biophysics Core Facility of the BMC for access to their instruments, helpful discussions and advice.

## Additional information

### Funding

| Funder | Grant reference number | Author |
|---|---|---|
| Deutsche Forschungsgemeinschaft | 263531414 / FOR2290 | Martin Zacharias Harald Steiner |

The funders had no role in study design, data collection and interpretation, or the decision to submit the work for publication.

### Author contributions

Lukas P Feilen, Conceptualization, Formal analysis, Investigation, Project administration, Visualization, Writing - review and editing; Shu-Yu Chen, Formal analysis, Investigation, Methodology, Validation, Visualization, Writing – original draft; Akio Fukumori, Construct generation, Resources, Writing - review and editing; Regina Feederle, Antibody generation, Resources; Martin Zacharias, Conceptualization, Funding acquisition, Resources, Supervision, Writing - review and editing; Harald Steiner, Conceptualization, Funding acquisition, Project administration, Supervision, Writing – original draft

### Author ORCIDs

Lukas P Feilen http://orcid.org/0000-0001-8221-6742
Regina Feederle http://orcid.org/0000-0002-3981-367X
Martin Zacharias http://orcid.org/0000-0001-5163-2663
Harald Steiner http://orcid.org/0000-0003-3935-0318

### Decision letter and Author response

Decision letter https://doi.org/10.7554/eLife.76090.sa1
Author response https://doi.org/10.7554/eLife.76090.sa2

## Additional files

### Supplementary files

• Transparent reporting form

### Data availability

For all figures the source data are provided in the respective source data files. The coordinate and trajectory files of all simulations can be accessed at Zenodo: https://doi.org/10.5281/zenodo.6487373.

The following previously published datasets were used:

| Author(s) | Year | Dataset title | Dataset URL | Database and Identifier |
| --- | --- | --- | --- | --- |
| Zhou R, Yang G, Guo X, Zhou Q, Lei J, Shi Y | 2019 | Recognition of the amyloid precursor protein by human γ-secretase | https://www.rcsb.org/structure/6IYC | RCSB Protein Data Bank, 6IYC |
| Li X, Dang S, Yan C, Gong X, Wang J, Shi Y | 2013 | Structure of a presenilin family intramembrane aspartate protease | https://www.rcsb.org/structure/4HYG | RCSB Protein Data Bank, 4HYG |

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

# Appendix 1

## Protein quality control

To check for the quality of the protein preparations, WT and mutant PSH were analyzed by DLS, BN-PAGE and nanoDSF. In DLS experiments, the Z-Average (Z-Ave) value for WT and mutant PSH was below 100 nm, which indicates that the different PSH preparations were not aggregated (*Appendix 1—figure 1A*). BN-PAGE immunoblot analysis showed a band for monomeric and dimeric PSH for all constructs but no higher molecular weight aggregate formation was observed (*Appendix 1—figure 1B*). nanoDSF experiments showed an inflection temperature ($T_i$) of round about 72.5 °C for all constructs (*Appendix 1—figure 1C*) indicating that the introduction of single point mutations into PSH does not influence the thermal stability and folding of the protein.

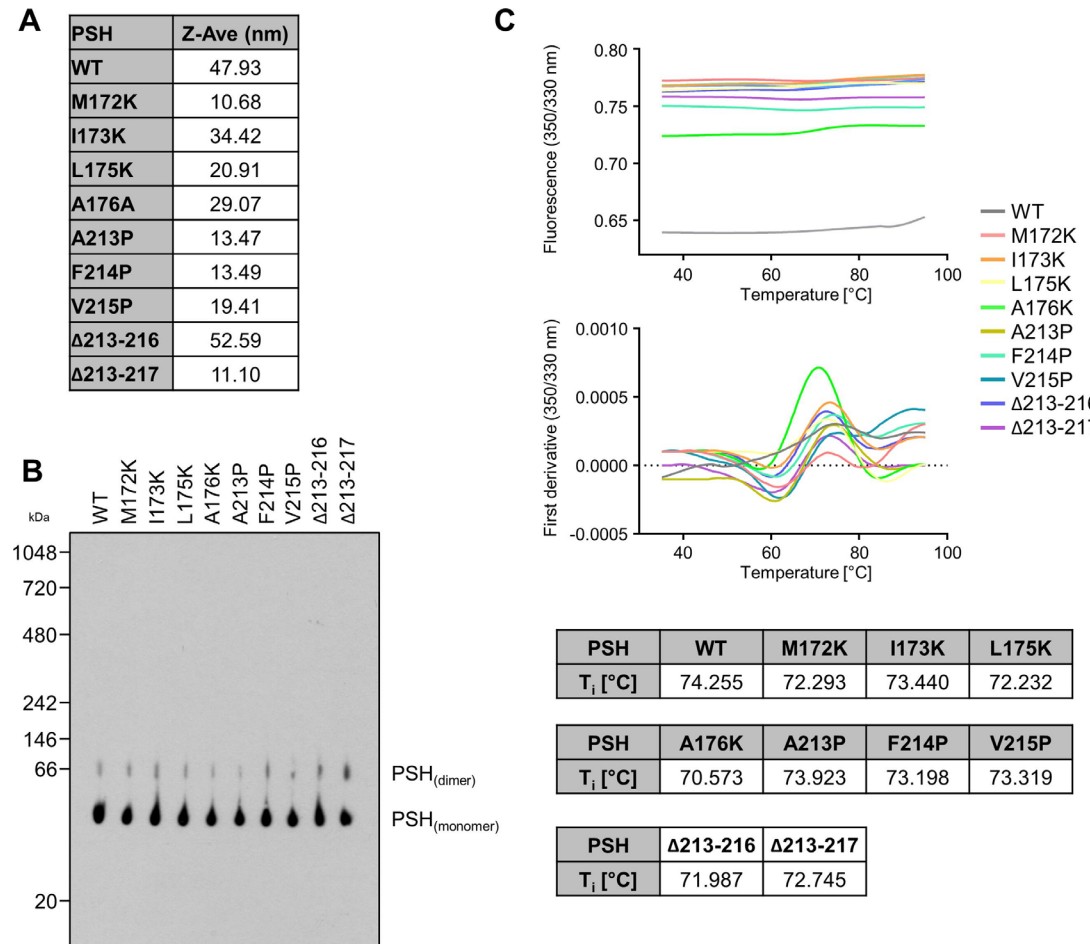

**A**

| PSH | Z-Ave (nm) |
|-----|-----------|
| WT | 47.93 |
| M172K | 10.68 |
| I173K | 34.42 |
| L175K | 20.91 |
| A176A | 29.07 |
| A213P | 13.47 |
| F214P | 13.49 |
| V215P | 19.41 |
| Δ213-216 | 52.59 |
| Δ213-217 | 11.10 |

| PSH | WT | M172K | I173K | L175K |
|-----|-----|-------|-------|-------|
| $T_i$ [°C] | 74.255 | 72.293 | 73.440 | 72.232 |

| PSH | A176K | A213P | F214P | V215P |
|-----|-------|-------|-------|-------|
| $T_i$ [°C] | 70.573 | 73.923 | 73.198 | 73.319 |

| PSH | Δ213-216 | Δ213-217 |
|-----|----------|----------|
| $T_i$ [°C] | 71.987 | 72.745 |

**Appendix 1—figure 1.** Quality control of WT and mutant PSH. (**A**) Analysis of protein aggregation of WT and mutant PSH by DLS. (**B**) Analysis of protein aggregation of WT and mutant PSH by BN-PAGE followed by immunoblotting for PSH (6F4). (**C**) Analysis of protein misfolding of WT and mutant PSH by nanoDSF.

