## [Editor Report]

This work provides a strong contribution to our understanding of intramembrane proteolysis and in particular the subtle structural but significant influence of the lipid bilayer on proteolytic activity and coordination of the active site geometry.

---

## [Decision Letter]

**Decision letter after peer review:**

[Editors’ note: the authors submitted for reconsideration following the decision after peer review. What follows is the decision letter after the first round of review.]

Thank you for submitting your work entitled "Active site geometry stabilization of a presenilin homolog by the lipid bilayer promotes intramembrane proteolysis" for consideration by *eLife*. Your article has been reviewed by 3 peer reviewers, including Joanne Lemieux as the Reviewing Editor and Reviewer #1, and the evaluation has been overseen by a Senior Editor. The following individual involved in review of your submission has agreed to reveal their identity: Michael S. Wolfe (Reviewer #3).

Our decision has been reached after a consultation among editors and reviewers. Based on these discussions and the individual reviews below, we regret to inform you that *eLife* will not be considering this submission for publication. We recognize the importance of this work and the progress made towards a better understanding of the factors that influence intramembrane proteolysis. Indeed, the reviewers' evaluation of the experimental work was positive overall. However, the reviewers also identified critical problems in the computational component. Although it seems possible to address these problems, it also seems that the revision would exceed the two-month period that is typically allowed. Hence, we believe the appropriate action is to reject the current version of your manuscript and allow you to consider your options. Should you decide to address all the concerns raised by the reviewers, we would be willing to examine a new version of your manuscript – but please note it would be considered a new submission, and therefore, that it might not be sent for review or be evaluated by the same reviewers.

*Reviewer #1:*

The activity of the archeal presenilin homolog (PSH) is assessed using a portion of the amyloid precursor substrate C99 substrate in DDM detergent and also in a POPC liposome environment. Mass spectrometry was used to precise indicate cleavage species as well as SDS. The results clearly show lipid influence both the rate of cleavage and the species generates. Modelling was used to determine structures of the PSH with the substrate in both app and holo forms. Similar to the cryoEM structures of PS with Notch and APP, a hybrid enzyme-substrate beta-sheet is observed between substrate and active site, suggesting a valid model for MD situations. These were used in MD simulations in both detergent and lipid environments. Importantly protonation states were analyzed for this Aspartyl protease, in which the two Asp residues were evaluated independently. D220 was determined to be the protonated Asp with D162 left charged in the simulations in different environments. The MD simulations show there were less structural changes in the DOPC embedded protein compared to the non-lipid sample, with a notable shift in TMD6a only in the DDM sample . MD simulations support the above experimentally observations that the lipid stabilize the E-S complex.

Overall this is an interesting paper that combines both experimental and structural studies to rationale differences in membrane environment and protease activity.

*Reviewer #2:*

I have several comments and suggestions for the authors to consider improving the quality of the paper:

1. Utilization of C100 for enzymatic analysis.

Authors utilized the recombinant C100-His6 as a substrate for the enzymatic assay. In contrast, they modeled the complex structure of PSH with C83, as the template cryo-EM structure was PS1-C83 complex. Essentially γ-secretase cleaves several substrates regardless the sequence. However, N-terminal length of the substrate might affect the proteolytic efficacy of the γ-secretase. In fact, Funamoto et al. (Nat Commun 2013) reported that AICD production from C83-FLAG was much greater than that from C99-FLAG (i.e., distinct Km and Vmax values). These data suggest the possibility that the formation of E-S complex is regulated by the N-terminal length of the substrate in the proteolytic mechanism of the γ-secretase. Thus, I would recommend the authors to compare the cleavage and trimming patterns of C83 by PSH in either DDM or POPC, as shown in the modeling analysis and the MD simulation. Such comparison strengthens the author's conclusion that the stabilization of E-S complex is critical to the intramembrane proteolysis.

2. Importance of TMD6a and hybrid β-sheet in the proteolysis by PSH.

Modeled structure of PSH-C83 strongly implicates the importance of these two structural elements in the proteolysis of PSH. However, it remains unclear whether these elements are truly required for the proteolytic activity of PSH. The possibility that these structures artificially modeled because the authors used the γ-secretase-C83 structure as a template. Thus, authors should test the proteolytic activity of mutant PSH that carries amino acid substitutions in TMD6a or beta2-strand to abolish the interaction with the substrate.

3. Figure presentation

In figure 5A, it is difficult to understand the difference of TMD6a conformation in DDM/POPC because of superimposed structure. They should be presented separately.

*Reviewer #3:*

In this manuscript, Feilen et al. combined biochemical experiments and molecular dynamics (MD) simulations to investigate the effects of detergent solubilization versus lipid vesicle reconstitution on intramembrane protease activity of an archaeal presenilin homolog (PSH). This PSH has been previously reported to process amyloid precursor (APP)-based substrate to amyloid β-peptides (A-β) in a manner closely similar to that accomplished by the presenilin-containing γ-secretase complex. The archaeal PSH is employed here as a surrogate to gain mechanistic insight into γ-secretase.

The authors showed that the carboxypeptidase-like activity of PSH is impaired in DDM micelles compared to a POPC lipid bilayer. Evidence is also provided suggesting that DDM-solubilized PSH binds more weakly to a transition-state analog inhibitor compared with POPC-bilayer PSH. Comparative MD simulations suggested that the lipid bilayer stabilized relevant structural elements of PSH for substrate binding (notably TMD6a) and formation of the enzyme active site geometry for proteolysis. A number of suggestions that could help improving the manuscript include the following:

1. While it is understood that archaeal PSH is taken as a surrogate for presenilin in the γ-secretase complex, the authors should explain why this is necessary. All the described experiments, including the MD simulations, could have been conducted with γ-secretase itself. In the discussion, the specific implications for γ-secretase (especially FAD mutations) should be de-emphasized and more emphasis put on the implications for intramembrane proteolysis in general.

2. For the MS analysis of A-β peptide products in Figure 1D, a table of observed vs. calculated m/z should be provided. Some of these peaks (A-β-43, -45, and -46) are quite weak. The same is true of AICD MS analysis in Figure 2D.

3. Given crystal structures of the PSH, is it more reasonable to preserve protein coordinates in these crystal structures, but only add the missing residues (e.g., TMD6a) using the PS1/γ-secretase cryo-EM structures as template? The Results section is vague about how the homology modeling of enzyme-substrate complex was generated; an additional sentence or two should be provided so the reader does not have to refer to the experimental section to answer this basic question.

4. Assuming pKa calculations depend on local geometry of the protonation site, the protein structure(s) used for the calculations need to be described clearly. Moreover, how do the residue pKa's depend on the protein structures (e..g, 4Y6K and 4HYG crystal structures of PSH, the apo and holo cryo-EM structures of PS1/γ-secretase, and simulation equilibrated protein structures in notably two different conformations with the D162-D220 distance centered around ~6.5 and ~8.2 Angstroms in Figure 6A).

5. It is unclear why the Amber force field was used in simulations of holo PSH with two different protonation states, but CHARMM36m in simulations of the apo and holo PSH in different membrane environments. It would help to evaluate the force field differences and potential effects by adding simulations using CHARMM36m to the holo PSH with different protonation states or simulations using Amber to the apo and holo PSH in different membrane environments.

6. In Figure 3A, what is the RMSD between the PSH homology model and its crystal structures?

7. In Figures 4C-4E, it would help to add error bars (standard deviations) to examine what differences are significant. The authors calculated RMSFs of PSH in the apo and holo forms to describe its stability in different lipid environments. It is better to show error bars (with total simulation times across different replicates) to describe RMSF differences in notably, the TMD6a, TMD4 and TM2-TM3 loop. There is a notable RMSF difference just beyond TMD6a, which should be mentioned and explained. In addition to RMSF, further simulation analysis such as comparison of the distances between the catalytic aspartate and scissile peptide bond in C83 could provide more insights.

8. In Figure 5A, it would help to quantitatively calculate and plot the helicity of TMD6a and/or secondary structures of residues in TMD6a as a function of simulation time and compare these quantities between the different simulated systems.

9. In Figure 5C-5D, the authors described that the DDM molecule can insert itself between TMD2 and TMD6 and intervenes intra- and intermolecular interactions and thus destabilize TMD6a. It would be better to have more explanation if these insertions are with just one lipid molecule or there are multiple molecules involved during different time frames of the simulations. In addition, it would be more convincing to see atomic detailed interaction between the DDM molecule and the protein, especially because DDM is a nonionic detergent. Moreover, it could help to calculate -SCD order parameters that are usually obtained from NMR experiments to measure orientational anisotropy of the C-H bonds lipid chains and quantify differences of the lipid orientations.

10. For Figure 6A, it would help to plot the D162-D220 distance vs. time in simulations of the different systems also. What are the corresponding distance values in the PSH crystal and PS1 cryo-EM structures? In addition to a main peak at ~6.5 Å distance, there seems another peak at ~8.2 Å distance between D162-D220; what is this conformational state?

11. In Figure 6B, there is substantial non-specific binding of the biotinylated Merck C to PSH in DDM; the parent inhibitor is essentially not competing, even at 20 microM. This makes the interpretation that specific binding is stronger in POPC vesicles less convincing. Some comment to this effect should be added to the Results section.

*Reviewer #4:*

Feilen et al. address an interesting mechanistic and biophysical question, namely the significance of the lipid environment for intramembrane enzymatic activity – through biochemical experiments and MD simulations. While the experimental component seems compelling, my opinion is that the structural/computational element is unsuitable for publication in *eLife*. It is well known that the construction of homology models relies on multiple arbitrary decisions, from the choice of template structure to the sequence alignments to the scoring function – which together imply a degree of inaccuracy that is simply unknown a priori. When a complex is modeled, the uncertainties accumulate. At the same time, MD simulations are, by design, highly sensitive to the details of the input structure. It follows, that systematic inaccuracies in the input model for an MD simulation might result in observations that have no mechanistic significance. At the level of an *eLife* publication, therefore, it is essential that the authors demonstrate that their conclusions are robust and independent from those built-in uncertainties. A possible way forward would be to carry out simulations of existing experimental structures that might be relevant (with or without minor modifications). Alternatively, or in addition, the authors could consider equally plausible but meaningfully different homology models, constructed on the basis of different assumptions. Either way, I do not believe the manuscript can move forward to publication without an extensive overhaul of the computational section (or its removal), so as to clearly ascertain the conclusions are indeed significant and robust.

A related issue pertains to the comparison of PC bilayers vs DDM micelles. The authors construct and examine one micelle system with a specific number of DDM molecules solubilizing the protein in a specific volume. This choice seems again arbitrary – or at least it is not explained. While the characteristics of lipid bilayer models in odellingn have been extensively studied and optimized (area per lipid, bending modulus, etc), I am unclear the same applies to DDM micelles. What are the observables that give the authors confidence that the structural and elastic properties of their micelle model are realistic? Given that the differences between the POPC and DDM simulations are ultimately modest, this comparative analysis needs to be much more systematic than it currently is to merit publication in *eLife*. As with the homology odelling, the question is whether different micelles or different DDM models would lead to alternative conclusions.

[Editors’ note: further revisions were suggested prior to acceptance, as described below.]

Thank you for resubmitting your work entitled “Active site geometry stabilization of a presenilin homolog by the lipid bilayer promotes intramembrane proteolysis” for further consideration by *eLife*. Your revised article has been evaluated by 3 reviewers, including Joanne Lemieux as the Reviewing Editor and reviewer #1 and overseen by José Faraldo-Gómez as Senior Editor.

Reviewers and editors agree that the manuscript has been improved but there are some remaining issues that need to be addressed, outlined below. While recognizing the merits of the work, all agree the authors at times overstate the novelty of their findings and that the narrative must be toned down on account of the published work in this area. Furthermore, the reviewers add some insight that could be brought into the discussion.

*Reviewer #1:*

The authors provide an exciting body of work to support the role of the lipid bilayer in stabilizing the coordination of active site residues. The PSH protein is a suitable model since it does not contain any co-factors and is an active protease on its own. The authors demonstrate that the lipid bilayer enhances processivity of PSH. The data is supported by in vitro cleavage assays in detergent and lipid vesicles. Furthermore, in this revised version there are MD analyses conducted of several apo- and substrate: protease complexes, which provide insight into the protease dynamics and interactions with substrates. The revisions add depth to the paper and is suitable for publication.

*Reviewer #2:*

Comments and suggestions that could improve the manuscripts are below:

1. In the case of the deletions/proline mutations in the hybrid β-sheet and the lysine mutations in TMD6a, I would have liked to see some type of experiment (DLS, SEC, etc.) to show that the mutations did not simply result in aggregated or misfolded protein. I find it troublesome that it is rare to see confirmation that point mutations to key residues did not completely disrupt protein folding, particularly when detergents are being used for solubilization.

2. It would be worthwhile to consider the possibility that the inhibitors partition into the DDM micelle in the detergent environment. This is a known phenomenon and could explain the poor inhibition of PSH by both L-685,458 and Merck C in the DDM environment, particularly if the DDM that was added to the POPC experiments was to aid in the solubilization of what I gather is a highly insoluble molecule. In general, more detail in the experimental section regarding the in vitro cleavage assay and affinity precipitations would be helpful.

3. The manuscript suggests that it is remarkable to see a rise in processivity when PSH is reconstituted in POPC membranes. In fact, highly inconsistent enzymatic activities (and structures!) have been observed for numerous enzymes in detergent vs. lipid bilayers (e.g. MsbA, MalFGK). It could be beneficial to consider these examples in the discussion and tone down the language, as it is not particularly surprising to see this sort of inconsistency in these different environments.

4. Both the in vitro and in silico experiments in the lipid environment were carried out in POPC, but PSH is an archaeal homologue. The lipids found in archaea are very different to other membranes, and while it may be outside the scope of this study to carry out the experiments in the natural PSH lipid environment (or not possible due to availability), it may be worthwhile running MD simulations a more representative environment, particularly because it well established that the identity of the annular lipids around an enzyme can significantly affect its activity.

5. Related to the above point, I realize that working with a single lipid simplifies things, but could or should the experiments in lipid bilayer not have been done in a brain lipid extract to more accurately recapitulate the environment of the enzyme that PSH is meant to be a surrogate for? Or as above, at the very least the MD simulations could have been done in a more representative environment.

*Reviewer #3:*

The manuscript by Feilen et al. underwent significant revisions and addressed the reviewer's concerns adequately. The manuscript focuses on the importance of the lipid environment for intramembrane proteolysis. Before I list a number of only smaller suggestions for this manuscript, I would like to mention that while I fully understand that this is beyond the scope of the present manuscript, it would have been interesting if the authors could have enforced the aspect of direct lipid-enzyme interactions, about which very little is known. For example, could the authors deduce amino acids in PSH that are important for the interaction with POPC molecules, and mutate those? Or is the stabilizing effect of POPC solely conveyed through the self-ordering of membrane molecules?

---

## [Author Response]

[Editors’ note: the authors resubmitted a revised version of the paper for consideration. What follows is the authors’ response to the first round of review.]

Reviewer #2:I have several comments and suggestions for the authors to consider improving the quality of the paper:1. Utilization of C100 for enzymatic analysis.Authors utilized the recombinant C100-His6 as a substrate for the enzymatic assay. In contrast, they modeled the complex structure of PSH with C83, as the template cryo-EM structure was PS1-C83 complex. Essentially γ-secretase cleaves several substrates regardless the sequence. However, N-terminal length of the substrate might affect the proteolytic efficacy of the γ-secretase. In fact, Funamoto et al. (Nat Commun 2013) reported that AICD production from C83-FLAG was much greater than that from C99-FLAG (i.e., distinct Km and Vmax values). These data suggest the possibility that the formation of E-S complex is regulated by the N-terminal length of the substrate in the proteolytic mechanism of the γ-secretase. Thus, I would recommend the authors to compare the cleavage and trimming patterns of C83 by PSH in either DDM or POPC, as shown in the modeling analysis and the MD simulation. Such comparison strengthens the author's conclusion that the stabilization of E-S complex is critical to the intramembrane proteolysis.

We thank the reviewer for the recommendations on performing cleavage assays with APP C83. In the new version of the manuscript, we now also analyzed the cleavage of C83 by PSH in the DDM and POPC environments to test whether substrate N-terminus length might affect our results. As shown in Figure 3A, we observed robust cleavage of C83 also by PSH in both environments. In line with our observations for C99 processing by PSH in the different environments, POPC also increased the processive cleavage of C83 resulting in the production of shorter p3 species (Figure 3B and Figure 3 – source data 1). The similarity of the C83 (Figure 3B and Figure 3 – source data 1) and C99 processivity data (Figure 3C and Figure 3 – source data 1) further strengthen our conclusion that the stabilization of the E-S complex is indeed critical for intramembrane proteolysis. Furthermore, these data inform that the differences in the processivity of PSH observed for C99 in detergent micelle vs lipid bilayer environments should be explorable at the structural level with models of a PSH–C83 E-S complex.

2. Importance of TMD6a and hybrid β-sheet in the proteolysis by PSH.Modeled structure of PSH-C83 strongly implicates the importance of these two structural elements in the proteolysis of PSH. However, it remains unclear whether these elements are truly required for the proteolytic activity of PSH. The possibility that these structures artificially modeled because the authors used the γ-secretase-C83 structure as a template. Thus, authors should test the proteolytic activity of mutant PSH that carries amino acid substitutions in TMD6a or beta2-strand to abolish the interaction with the substrate.

We agree with the reviewer that the two important structural elements TMD6a and 2-strand might have arisen from the PS1 template of our homology modeling and therefore might be only structural artefacts.

Therefore, we performed mutational analysis of these structural elements and assed their proteolytic activity. As shown in Figure 4E and Figure 7F, mutations within TMD6a and 2-strand abolished the activity of PSH towards C99. Therefore we believe that both structural elements are functionally present in PSH and do not represent an artefact from model building.

3. Figure presentationIn figure 5A, it is difficult to understand the difference of TMD6a conformation in DDM/POPC because of superimposed structure. They should be presented separately.

We agree with the reviewer. In our new version of the manuscript, we now provide in Figure 6 separate figures for the PSH structures in POPC (Figure 6A) and DDM (Figure 6B-D). This makes it easier to differ between the stable TMD6a conformation in POPC and the unfolded conformation in DDM.

Reviewer #3:In this manuscript, Feilen et al. combined biochemical experiments and molecular dynamics (MD) simulations to investigate the effects of detergent solubilization versus lipid vesicle reconstitution on intramembrane protease activity of an archaeal presenilin homolog (PSH). This PSH has been previously reported to process amyloid precursor (APP)-based substrate to amyloid β-peptides (A-β) in a manner closely similar to that accomplished by the presenilin-containing γ-secretase complex. The archaeal PSH is employed here as a surrogate to gain mechanistic insight into γ-secretase.The authors showed that the carboxypeptidase-like activity of PSH is impaired in DDM micelles compared to a POPC lipid bilayer. Evidence is also provided suggesting that DDM-solubilized PSH binds more weakly to a transition-state analog inhibitor compared with POPC-bilayer PSH. Comparative MD simulations suggested that the lipid bilayer stabilized relevant structural elements of PSH for substrate binding (notably TMD6a) and formation of the enzyme active site geometry for proteolysis. A number of suggestions that could help improving the manuscript include the following:1. While it is understood that archaeal PSH is taken as a surrogate for presenilin in the γ-secretase complex, the authors should explain why this is necessary. All the described experiments, including the MD simulations, could have been conducted with γ-secretase itself. In the discussion, the specific implications for γ-secretase (especially FAD mutations) should be de-emphasized and more emphasis put on the implications for intramembrane proteolysis in general.

As recommended by the reviewer, we have now more clearly described in the introduction why PSH is needed as presenilin/-secretase surrogate for our study. The principal problem is that -secretase, unlike PSH, is not active without lipids (e.g., Zhou et al. 2010), so that a comparison of processivity in lipid bilayer versus detergent-micelle environment is not possible. This type of analysis can therefore not be done for -secretase itself. As further recommended, we also followed the reviewer´s suggestion to de-emphasize the implications for FAD mutations in -secretase and rewrote the respective part in the Discussion, highlighting more the implications of our findings for intramembrane proteolysis in general.

References:

Zhou et al. (2010) Dependency of -secretase complex activity on the structural integrity of the bilayer, Biochem. Biophys. Res. Commun., 12, 402(2), 291-296.

2. For the MS analysis of A-β peptide products in Figure 1D, a table of observed vs. calculated m/z should be provided. Some of these peaks (A-β-43, -45, and -46) are quite weak. The same is true of AICD MS analysis in Figure 2D.

As requested by the reviewer, we now provide the calculated and observed masses for the peaks of our MS analysis. These are found in the corresponding source data files (Figure 1 – source data 1 and Figure 2 – source data 1). We have worked hard to get better spectra for the weaker peaks but were so far not successful and could thus not replace this figure with better quality data. Besides matching masses, the peaks are specific as demonstrated by their absence in inhibitor controls (Figure 1 —figure supplement 1 and Figure 2 —figure supplement 2).

3. Given crystal structures of the PSH, is it more reasonable to preserve protein coordinates in these crystal structures, but only add the missing residues (e.g., TMD6a) using the PS1/γ-secretase cryo-EM structures as template? The Results section is vague about how the homology modeling of enzyme-substrate complex was generated; an additional sentence or two should be provided so the reader does not have to refer to the experimental section to answer this basic question.

We thank the reviewer for his recommendations on model building and also agree that they should be better described in the Results section. In the new version of the manuscript, for which two additional C83-bound PSH models were generated, this is now more clearly described in the main text. The models are generated by three different approaches, including one considering only the PS1 cryo-EM structure (model 1, Table 1) and two including also the coordinates of the PSH crystal structure (model 2 and model 3, Table 1). It is important to note that although the X-ray crystal structure of apo PSH is available (PDB 4HYG), residues 182 to 209 had been proteolytically removed and five stabilizing mutations were introduced. Furthermore, due to its tetrameric nature, the possible substrate entries between TMDs 2 and 6 and TMDs 2 and 3 are blocked implicating an artificial TMD arrangement in the crystal structure. Therefore, it is questionable if the 4HYG structure alone is a good starting model for a PSH-substrate complex. However, we like to emphasize that our model 2 (corresponding to the model used in the first version of the manuscript) is indeed based mostly on the 4HYG structure, but the missing relevant parts for substrate binding are replaced by the PS1 template (Table 1). Our third model considers also parts of the structural information of PS1/-secretase cryo-EM structures ( Table 1), aiming to generate a more accurate model that contains the reliable part of the apo-PSH structure and the critical features revealed in the PS1-C83 complex. As stated above, we also include a more detailed description of the homology modeling in the new version of the manuscript in the Results and Materials and methods sections.

4. Assuming pKa calculations depend on local geometry of the protonation site, the protein structure(s) used for the calculations need to be described clearly. Moreover, how do the residue pKa's depend on the protein structures (e..g, 4Y6K and 4HYG crystal structures of PSH, the apo and holo cryo-EM structures of PS1/γ-secretase, and simulation equilibrated protein structures in notably two different conformations with the D162-D220 distance centered around ~6.5 and ~8.2 Angstroms in Figure 6A).

We agree and have accordingly revised the relevant sections in our new manuscript providing additional information regarding the pKa values of the catalytic residues. The pKa prediction on the two aspartic acids of PS1/PSH are now listed in Table 2. For basically all available structures (except for PDB 5FN5, a structure of one of the PS1/-secretase apo-states (apo-state 3)), D220 of PSH (D385 in PS1) in TMD7 is predicted to have a higher pKa than the second aspartic acid residue in TMD6, D162 (D257 in PS1).

5. It is unclear why the Amber force field was used in simulations of holo PSH with two different protonation states, but CHARMM36m in simulations of the apo and holo PSH in different membrane environments. It would help to evaluate the force field differences and potential effects by adding simulations using CHARMM36m to the holo PSH with different protonation states or simulations using Amber to the apo and holo PSH in different membrane environments.

In our previous computational work, the AMBER force field is usually implemented since it is more compatible with our computational engine. However, the AMBER force field does not have the parameters describing the DDM molecules, and therefore we needed to switch the force field from AMBER to CHARMM36m. Since the parameter files with the CHARMM36m force field generated by CHARMM-GUI server cannot be modified easily, changing the protonation state and calculating the substrate-binding energy using the MMPBSA method might include more uncertainties such as the need to define the implicit radii of each atom etc. The verification of the protonation states of the catalytic residues using the AMBER force field is now removed in the new manuscript and is replaced by showing the list of pKa predictions in Table 2.

6. In Figure 3A, what is the RMSD between the PSH homology model and its crystal structures?

We give now more details of the RMSD values in Figure 4 —figure supplement 1. We show the RMSD values between our three models and the two published PSH crystal structures (PDB 4HYG, 4Y6K, chain B) as well as the RMSD between the homology models that we generated.

7. In Figures 4C-4E, it would help to add error bars (standard deviations) to examine what differences are significant. The authors calculated RMSFs of PSH in the apo and holo forms to describe its stability in different lipid environments. It is better to show error bars (with total simulation times across different replicates) to describe RMSF differences in notably, the TMD6a, TMD4 and TM2-TM3 loop. There is a notable RMSF difference just beyond TMD6a, which should be mentioned and explained. In addition to RMSF, further simulation analysis such as comparison of the distances between the catalytic aspartate and scissile peptide bond in C83 could provide more insights.

As recommended by the reviewer, standard deviations of the RMSFs across three different replicates are now included in the plots as shaded areas (Figure 5C and E, Figure 5 —figure supplement 1B, Figure 5 —figure supplement 3B and C, Figure 6 —figure supplement 3D, F and G, Figure 7B and E). The RMSF difference between bilayer and micelle environments at the region immediately C-terminal of TMD6a (T182A184) are now mentioned and discussed in new version. Similar as what was observed for TMD6a, these residues are also interacting with the DDM molecules inserted between TMD3 and TMD4.

In Figure 8 —figure supplement 2, we provide a more detailed geometric picture of the active site of the two sampled conformations shown in Figure 8B including also distances of the catalytic residues to the substrate scissile bond. Although both conformations are capable of forming hydrogen bonds with the protonated aspartic acid (D220), the geometry with a larger C-C distance between the catalytic aspartic acids is less likely for the enzyme to execute the hydrolysis reaction because of the larger distance between the scissile bond and the unprotonated aspartic acid (D162). Overall, our additional analysis of various relevant distances shown in Figure 8 —figure supplement 2 indicates that the CC distance should be the preferred criterion to characterize a functional active site geometry as compared to merely using H-bond distances.

8. In Figure 5A, it would help to quantitatively calculate and plot the helicity of TMD6a and/or secondary structures of residues in TMD6a as a function of simulation time and compare these quantities between the different simulated systems.

As recommended by the reviewer, evaluation of the secondary structure of TMD6a and surrounding residues is now shown in Figure 5 —figure supplement 2 and Figure 7 —figure supplement 1.

9. In Figure 5C-5D, the authors described that the DDM molecule can insert itself between TMD2 and TMD6 and intervenes intra- and intermolecular interactions and thus destabilize TMD6a. It would be better to have more explanation if these insertions are with just one lipid molecule or there are multiple molecules involved during different time frames of the simulations. In addition, it would be more convincing to see atomic detailed interaction between the DDM molecule and the protein, especially because DDM is a nonionic detergent. Moreover, it could help to calculate -SCD order parameters that are usually obtained from NMR experiments to measure orientational anisotropy of the C-H bonds lipid chains and quantify differences of the lipid orientations.

As recommended by the reviewer, the current manuscript now describes in more detail how DDM perturbs intramolecular PSH TMD interactions. Figure 6B and C shows that the inserted DDM molecule disturbs the TMD6a helix mainly by unspecific hydrogen bond interactions with the PSH backbone. This also happens on the residues immediately Cterminal to TMD6a (Figure 6D). During our simulation time, mostly only one DDM molecule was found to insert into the gap and interfere with the PSH backbone. As further recommended, we calculated lipid chain C-H bond order parameters to quantify differences of the lipid orientation. The S_CH_ order parameters calculated are now included in Figure 6 —figure supplement 2B and demonstrate the average orientation of DDM and POPC molecules.

10. For Figure 6A, it would help to plot the D162-D220 distance vs. time in simulations of the different systems also. What are the corresponding distance values in the PSH crystal and PS1 cryo-EM structures? In addition to a main peak at ~6.5 Å distance, there seems another peak at ~8.2 Å distance between D162-D220; what is this conformational state?

As recommended by the reviewer, the D162-D220 C-C distance over simulation time is now shown in Figure 8 —figure supplement 1C. The C-C distances of the substrate-bound structures are not measurable since D385 is mutated to alanine. The C-C distances in other PDB structures of PSH and PS1/-secretase are 3.99 Å (4HYG), 3.99 Å (4UIS), 6.58 Å (5A63), 5.35 Å (5FN5), 11.48 Å (5FN4), 5.06 Å (5FN3), and 3.89 Å (5FN2). The D162-D220 distance at ~8.2 Å corresponds to the geometry where more than two water molecules are accommodated in the catalytic center.

11. In Figure 6B, there is substantial non-specific binding of the biotinylated Merck C to PSH in DDM; the parent inhibitor is essentially not competing, even at 20 microM. This makes the interpretation that specific binding is stronger in POPC vesicles less convincing. Some comment to this effect should be added to the Results section.

This effect puzzled us as well. We agree with the suggestion of the reviewer and have thus commented on this effect regarding its interpretation in the results. As seen in the calculations of binding presented in Figure 8 – source data 1, binding of PSH is, however, still inhibited by the parental compound L-685,458 in DDM, although only to a very minor extent and much less compared to POPC. There was generally also much stronger background binding to the streptavidin beads in the absence of Merck C in DDM, adding to the low amount of specifically captured PSH in this condition. We thus speculate that the more labile active site in DDM also weakens the competition of binding. The reduced levels of specific binding in DDM support the interpretation that the active site geometry is more stabilized in the POPC bilayer. In support of this view, additional enzyme inhibition experiments presented in Figure 8E and F showed that L-685,458 and Merck C both inhibit PSH less well in DDM than in POPC.

Reviewer #4:Feilen et al. address an interesting mechanistic and biophysical question, namely the significance of the lipid environment for intramembrane enzymatic activity – through biochemical experiments and MD simulations. While the experimental component seems compelling, my opinion is that the structural/computational element is unsuitable for publication in eLife. It is well known that the construction of homology models relies on multiple arbitrary decisions, from the choice of template structure to the sequence alignments to the scoring function – which together imply a degree of inaccuracy that is simply unknown a priori. When a complex is modeled, the uncertainties accumulate. At the same time, MD simulations are, by design, highly sensitive to the details of the input structure. It follows, that systematic inaccuracies in the input model for an MD simulation might result in observations that have no mechanistic significance. At the level of an eLife publication, therefore, it is essential that the authors demonstrate that their conclusions are robust and independent from those built-in uncertainties. A possible way forward would be to carry out simulations of existing experimental structures that might be relevant (with or without minor modifications). Alternatively, or in addition, the authors could consider equally plausible but meaningfully different homology models, constructed on the basis of different assumptions. Either way, I do not believe the manuscript can move forward to publication without an extensive overhaul of the computational section (or its removal), so as to clearly ascertain the conclusions are indeed significant and robust.

The reviewer´s concerns on using comparative modeling are well taken – we are aware of the mentioned limitations and the uncertainties that can arise. As recommended by the reviewer, we have thus generated two additional model structures (Figure 4 —figure supplement 1) in the new version based on the same sequence alignment method but different in the ways of including different structural information from the existing structures (Figure 4 —figure supplement 2 and Table 1). It is important to note that although the X-ray crystal structure of apo PSH is available (PDB 4HYG), residues 182 to 209 are missing and five mutations are introduced. Furthermore, the tetrameric complex of four PSH molecules leads to a crystallographic artefact in the unit cell, which might block possible substrate entry sites between TMDs 2 and 6 and TMDs 2 and 3. Therefore, the available crystal structure is presumably far away from the substrate-bound structure. Our homology models consider also the structural information of PS1/-secretase cryo-EM structures in complex with the C83 substrate (Table 1), aiming to generate a more accurate model that contains the reliable part of apo PSH structure and the critical features revealed in the PS1-C83 complex.

We then performed comparative simulations on all model structures and based on the results argue that model 2 (which contains the reliable part of apo PSH and the substrate-binding region modelled based on the PS1 substrate-bound structure) is most realistic. Since our MD simulation studies on the model structures gave consistent results, we are confident that give likely and reliable explanations for the experimental results.

In addition, simulations can be used to create a hypothesis or to make predictions on the role of residues or protein segments. In this regard, our MD simulations are quite successful: based on the computational data, biochemical experiments were performed to evaluate the importance of the TMD6a and 2 structural features that were identified for holo PSH. By mutational analysis, we could indeed show that TMD6a and 2-strand are critical for substrate cleavage (Figure 4E and Figure 7F), strongly suggesting the reliability of homology model 2, which was selected as our working model. Therefore, we believe that our homology-modeling approach is suitable to study the dynamics of PSH embedded in two different environments at an atomic scale.

Taken together, we thus believe that the combination of our biochemical and computational approaches provides us with a picture of not only the unprecedented substrate-bound PSH structure but also with a molecular explanation of how a micelle environment destabilizes the active site geometry and interrupts the stability of the E-S complex.

A related issue pertains to the comparison of PC bilayers vs DDM micelles. The authors construct and examine one micelle system with a specific number of DDM molecules solubilizing the protein in a specific volume. This choice seems again arbitrary – or at least it is not explained. While the characteristics of lipid bilayer models in odellingn have been extensively studied and optimized (area per lipid, bending modulus, etc), I am unclear the same applies to DDM micelles. What are the observables that give the authors confidence that the structural and elastic properties of their micelle model are realistic? Given that the differences between the POPC and DDM simulations are ultimately modest, this comparative analysis needs to be much more systematic than it currently is to merit publication in eLife. As with the homology odelling, the question is whether different micelles or different DDM models would lead to alternative conclusions.

We agree with the reviewer that the chosen DDM micelle configuration could potentially be another concern and have thus followed the advice to control our data for such micelle-mediated effects. As recommended by the reviewer, our new manuscript now also includes an additional simulation system with 50% more DDM molecules (Figure 6 —figure supplement 3) The number of DDM molecules and micelle size were chosen based on Cheng et al. 2013, who parameterized the micelle parameters. By comparing the size of PSH to the size of proteins OmpA (PDB 1BXW) and OmpF (PDB 3POX) simulated by Cheng et al., our substrate-bound PSH is larger than OmpA but smaller than the OmpF trimer. With 80 DPC (n-dodecylphosphocholine) molecules in the former system and 160 in the latter, we consider 150 DDM molecules being a suitable number to investigate its dynamics in our system. In addition, the dynamics of the newly constructed holo PSH in the larger DDM micelle is consistent with the initial micelle size, which further consolidates the impact of DDM shown in our study. In the new manuscript version, we also include the area per lipid of POPC molecules (Figure 6 —figure supplement 2C) as well as S_CH_ order parameters of POPC and DDM (Figure 6 —figure supplement 2B). Overall, the values of area per lipid and the S_CH_ value for POPC agree well with the experimental data. Although the S_CH_ order parameter for DDM is to our knowledge not available to date and was therefore computed from our simulations, our analysis shows how disordered the DDM molecules are and from our structural depiction, we could illustrate how the disordered character of DDM perturbs the intramolecular interactions of PSH, in particular with those of TMD6a.

In conclusion, while we acknowledge the possibility that different sizes of micelle or lipid bilayer models may give divergent results from our initially chosen simulation systems, our new data with a larger DDM micelle size now suggest that this is rather unlikely as our principal observations were recapitulated with the additional analyses of alternative detergent micelle configurations. These additional data are thus overall in line with what one would expect for these two different conditions as the basic geometrical properties of a micelle, a spherical amphiphilic environment with disordered detergent molecules, and those of a bilayer, a relatively flat environment with ordered phospholipids molecules, are principally also shared by other detergent or phospholipid molecules.

References:

Cheng et al. (2013) CHARMM-GUI micelle builder for pure/mixed micelle and protein/micelle complex systems, J. Chem. Inf. Model, 53(8), 2171-

2180.

[Editors’ note: what follows is the authors’ response to the second round of review.]

Reviewers and editors agree that the manuscript has been improved but there are some remaining issues that need to be addressed, outlined below. While recognizing the merits of the work, all agree the authors at times overstate the novelty of their findings and that the narrative must be toned down on account of the published work in this area. Furthermore, the reviewers add some insight that could be brought into the discussion.Reviewer #2:Comments and suggestions that could improve the manuscripts are below:1. In the case of the deletions/proline mutations in the hybrid β-sheet and the lysine mutations in TMD6a, I would have liked to see some type of experiment (DLS, SEC, etc.) to show that the mutations did not simply result in aggregated or misfolded protein. I find it troublesome that it is rare to see confirmation that point mutations to key residues did not completely disrupt protein folding, particularly when detergents are being used for solubilization.

We agree with the reviewer that mutations might potentially induce aggregation and/or misfolding of the protein. To rule this out, we performed dynamic light scattering (DLS), Blue Native PAGE (BN-PAGE) and nano differential scanning fluorometry (nanoDSF) experiments with WT and mutant PSH. As shown in Appendix 1 – figure 1 A (DLS), B (BN-PAGE) and C (nanoDSF), all studied mutants behaved similar as WT PSH in these analyses suggesting that the mutants do not disrupt protein folding.

2. It would be worthwhile to consider the possibility that the inhibitors partition into the DDM micelle in the detergent environment. This is a known phenomenon and could explain the poor inhibition of PSH by both L-685,458 and Merck C in the DDM environment, particularly if the DDM that was added to the POPC experiments was to aid in the solubilization of what I gather is a highly insoluble molecule. In general, more detail in the experimental section regarding the in vitro cleavage assay and affinity precipitations would be helpful.

We thank the reviewer for pointing out this possibility, which we took into consideration.

However, in the POPC environment, DDM was not added to aid the solubilization of the L-685,458 γ-secretase inhibitor (which is added to the assays from a stock solution in DMSO), but to ensure enzyme activity of PSH. As we show in Author response image 1, PSH is not active in the POPC bilayer without the addition of small amounts of DDM (below the CMC). Interestingly, this seems to be a general phenomenon for presenilin-type intramembrane proteases and was already reported for γ-secretase in our Winkler et al. 2012 study. As shown in Supplementary figure 2B of this paper, also γ-secretase requires the addition of small amounts of CHAPSO detergent for activity when it is reconstituted in POPC vesicles. The reason for this requirement is however not understood yet. We included this information in the Materials and methods in our revised version and as suggested by the reviewer have also slightly revised the relevant text passages of our cleavage and inhibitor binding assays to provide the reader with more detail.

**Author response image 1. sa2fig1:** Comparison of PSH activity in the POPC bilayer with and without the addition of DDM. Analysis of PSH activity in POPC vesicles with and without the addition of 0.008% DDM after incubation with C100-His_6_ at 37 °C overnight by immunoblotting for AICD (Y188) and Aβ (2D8). The asterisks mark substrate degradation bands, which are independent of PSH cleavage.

Furthermore, while we cannot completely rule out the possibility of some inhibitor partition into DDM micelles, we do think that the inhibitors reach the active site in sufficient amounts due to their principal chemical properties. The mimicking of the transition state of a substrate peptide bond attacked by a presenilin-type aspartyl protease should direct these inhibitors to the active site. In fact, structural investigations of PSH with the γ-secretase inhibitor L-682,679, which is from the same chemical class as L-685,458 and Merck C, showed that the inhibitor enters the catalytical cleft (Dang et al. 2015). Also, for γ-secretase it was observed that L-685,458 binds directly at the active site (Yang et al. 2021). In addition, in MD simulations of γ-secretase with L-685,458 the inhibitor stays tightly bound to the active site (Hitzenberger et al. 2019).

References:

Winkler et al. (2012) Generation of Alzheimer disease-associated amyloid β42/43 peptide by γ-secretase can be inhibited directly by modulation of membrane thickness, J. Biol. Chem., 287(25), 21326-21334.

Dang et al. (2015) Cleavage of amyloid precursor protein by an archaeal presenilin homologue PSH, Proc. Natl. Acad. Sci. USA, 112(11), 3344-3349.

Yang et al. (2021) Structural basis of γ-secretase inhibition and modulation by small molecule drugs, Cell, 184(2), 521-533.e14.

Hitzenberger et al. (2019) Uncovering the binding mode of γ-secretase inhibitors, ACS Chem. Neurosci., 10(8), 3398-3403.

3. The manuscript suggests that it is remarkable to see a rise in processivity when PSH is reconstituted in POPC membranes. In fact, highly inconsistent enzymatic activities (and structures!) have been observed for numerous enzymes in detergent vs. lipid bilayers (e.g. MsbA, MalFGK). It could be beneficial to consider these examples in the discussion and tone down the language, as it is not particularly surprising to see this sort of inconsistency in these different environments.

We thank the reviewer for drawing our attention to studies on other membrane proteins that, for various reasons, show distinct changes in their activities in detergent micelles vs. membrane bilayers. However, in our study, we could consistently link activity differences in these two environments to structural changes of the enzyme-substrate complex and the active site geometry of the protease as determined by MD simulations that were in agreement with the biochemical analyses which included mutational analysis as well as the use of inhibitors probing the conformation of the active site. Nevertheless, since our findings may be misconceived in the light of studies with other membrane proteins in different environments, we followed the advice of the reviewer and toned down our wording by changing two sentences of the manuscript in the summarizing paragraph of the Discussion. This section is now modified as follows:

“Taken together, in good correlation between experimental and simulation data, our results with PSH as a model intramembrane protease highlight an important role of the membrane lipid environment in providing a stabilized E-S conformation that is crucial for substrate processing in intramembrane proteolysis.”

This rewording should capture the novelty of the insights provided by our study without overstatements.

4. Both the in vitro and in silico experiments in the lipid environment were carried out in POPC, but PSH is an archaeal homologue. The lipids found in archaea are very different to other membranes, and while it may be outside the scope of this study to carry out the experiments in the natural PSH lipid environment (or not possible due to availability), it may be worthwhile running MD simulations a more representative environment, particularly because it well established that the identity of the annular lipids around an enzyme can significantly affect its activity.

As already noted by the reviewer, archaeal lipids are not available for biochemical experiments, and we can therefore not perform experiments with the natural lipids for this protease. Additionally, to our knowledge nothing is known about the lipid composition of *Methanoculleus marisnigri*, the archaeal origin of PSH. Therefore, it will be difficult to perform biochemical experiments in a natural PSH environment and given this limitation, accompanying MD simulations will have only little if any informative value. Additionally, proper force fields for archaeal lipids are only limitedly available. Furthermore, the study represented here uses a non-natural substrate of PSH, since so far, no natural substrates are described. Even though we agree with the reviewer that the identity of the annular lipids around an enzyme can significantly affect its activity, we think that investigating PSH in a natural lipid environment would in this case also best be done with a natural substrate in order to study a homologous system. But we agree that while such a study is currently not doable and also outside the scope of this present study, it could be an interesting subject of future studies.

5. Related to the above point, I realize that working with a single lipid simplifies things, but could or should the experiments in lipid bilayer not have been done in a brain lipid extract to more accurately recapitulate the environment of the enzyme that PSH is meant to be a surrogate for? Or as above, at the very least the MD simulations could have been done in a more representative environment.

Simulating PSH in a bilayer composed of brain lipids is challenging due to the complex composition of brain lipids. From the available brain lipid extracts – also used in our experiments – only 40% of the lipids are known. This makes an accurate modeling of a brain lipid environment impossible. Furthermore, in our timeframe of 600 ns it will not be possible to sample all possibilities in a complex lipid system and influences of low abundance lipids might not be recognized at all. Nevertheless, we performed the cleavage assay with PSH reconstituted in vesicles prepared from a brain lipid extract and observed that this complex lipid environment also facilitates the processive cleavage of PSH (Author response image 2) indicating that this observation is not limited to the lipid type used in our studies.

**Author response image 2. sa2fig2:** Cleavage of C99 by PSH reconstituted in brain lipid vesicles. (A) Analysis of PSH activity in DDM micelles, POPC vesicles and brain lipid vesicles after incubation with C100-His_6_ substrate at 37 °C overnight by immunoblotting for AICD (Y188) and Aβ (2D8). The asterisks mark substrate degradation bands, which are independent of PSH cleavage. (B) Separation of Aβ species produced by PSH in DDM micelles, POPC vesicles and brain lipid vesicles by Tris-Bicine urea SDS-PAGE and analysis by immunoblotting for Aβ (2D8). (C) MALDI-TOF MS analysis of Aβ species generated by PSH in brain lipid vesicles at pH 7.0. The intensity of the highest peak was set to 100%.

Reviewer #3:The manuscript by Feilen et al. underwent significant revisions and addressed the reviewer's concerns adequately. The manuscript focuses on the importance of the lipid environment for intramembrane proteolysis. Before I list a number of only smaller suggestions for this manuscript, I would like to mention that while I fully understand that this is beyond the scope of the present manuscript, it would have been interesting if the authors could have enforced the aspect of direct lipid-enzyme interactions, about which very little is known. For example, could the authors deduce amino acids in PSH that are important for the interaction with POPC molecules, and mutate those? Or is the stabilizing effect of POPC solely conveyed through the self-ordering of membrane molecules?

We agree with the reviewer that it would be interesting to understand whether the effects we observed in the bilayer system are caused by a direct lipid-protein interaction or solely conveyed through the self-ordering of membrane molecules. Therefore, we analyzed the residue-wise solvent residence time of POPC and DDM using PyLipID (Song et al. 2022) with a dual cutoff distance of 1.2 Å and 0.8 Å (Author response image 3). Interestingly, this analysis shows that both POPC and DDM molecules reside longest in the gap between the Nterminal part of the TMD of C83 and the N-terminal part of TMD3 of PSH (Author response image 3, upper panel). Furthermore, POPC molecules show additional binding to several other regions (e.g. the loop between TMD7 and TMD8) (Author response image 3, lower panel). The long residence time between C83 and TMD3 of PSH is due to the large gap into which POPC and DDM can insert, whereas this entry is blocked by the hydrophobic interaction between I31 of C83 and I242 of nicastrin in the C83-bound γ-secretase complex (Author response image 3). Strikingly, the POPC molecule that we identified residing at the surface between TMD1, TMD5, TMD7 and TMD8 binds in a conformation similar to the ones resolved in the C83-bound and Notch1-bound γ-secretase complex structures (Author response image 3).

**Author response image 3. sa2fig3:** Binding of POPC and DDM molecules on the PSH surface. (A) The residue-wise averaged solvent residence time of POPC (blue) or DDM (red) on PSH surface. (B) Visualization of the DDM (middle) and POPC (right) residence time based on (A) from two different angles. Surfaces with long, medium and short lipid residence time are colored in blue, white, and red, respectively. Residence time is visualized in surface representation. (C) Binding of DDM (left) and POPC (middle) molecules in the gap between C83 and PSH TMD3 in the PSH simulations. In γ-secretase, this gap is blocked by the hydrophobic contact between nicastrin I242 (green) and C83 I31 (orange, right, 6IYC). (D)Binding of a POPC molecule to the PSH–C83 complex from MD simulations (left), Notchbound γ-secretase (middle, 6IDF) and C83-bound γ-secretase (right, 6IYC).

The similar lipid interaction sites in PSH and PS1 are very interesting and might strengthen the hypothesis that direct lipid-enzyme interactions might influence the enzyme activity. This is an interesting starting point for future studies on the lipid-enzyme interaction but is in our opinion out of scope for our present study. Of note, previous structural investigations in different intramembrane proteases have identified lipid molecules that were bound to the respective intramembrane protease. But none of these studies revealed a functional role of the enzymelipid interaction (Ben-Shem et al. 2007, Lemieux et al. 2007, Bai et al. 2015, Yang et al. 2019, Zhou et al. 2019).

References:

Song et al. (2022) PyLipID: A python package for analysis of protein-lipid interactions from molecular dynamics simulations. J. Chem. Theory Comput., 18(2), 1188-1201.

Ben-Shem et al. (2007) Structural basis for intramembrane proteolysis by rhomboid serine proteases, Proc. Natl. Acad. Sci. USA. 104(2), 462-466.

Lemieux et al. (2007) The crystal structure of the rhomboid peptidase from Haemophilus influenzae provides insight into intramembrane proteolysis, Proc. Natl. Acad. Sci. USA, 104(3), 750-754.

Bai et al. (2015) An atomic structure of human γ-secretase, Nature, 525(7568), 212-217.

Yang et al. (2019) Structural basis of Notch recognition by human γ-secretase, Nature, 565(7738), 192-197. Zhou et al., (2019) Recognition of the amyloid precursor protein by human γ-secretase, Science, 363(6428), eaaw0930.